# NUMERICAL MODELLING OF LANDSCAPE AND SEDIMENT FLUX RESPONSE TO PRECIPITATION RATE CHANGE

John J. Armitage[1], Alexander C. Whittaker[2], Mustapha Zakari[1], and Benjamin Campforts[3]

[1]Institut de Physique du Globe de Paris, Université Sorbonne Paris Cité, Paris, France
[2]Department of Earth Science and Engineering, Imperial College London, London, UK
[3]Division Geography, Department of Earth and Environmental Sciences, KU Leuven, Heverlee, Belgium

*Correspondence to:* John Armitage (armitage@ipgp.fr)

**Abstract.** Laboratory-scale experiments of erosion have demonstrated that landscapes have a natural (or intrinsic) response time to a change in precipitation rate. In the last few decades there has been a growth in the development of numerical models that attempt to capture landscape evolution over long time-scales. However, there is still an uncertainty over validity of the basic assumption of mass transport that are made in deriving these models. In this contribution we therefore return to a principle assumption of sediment transport within the mass balance for surface processes, and explore the sensitivity of the classic end-member landscape evolution models, and the sediment fluxes they produce, to a change in precipitation rates. One end-member model takes the mathematical form of a kinetic wave equation and is known as the stream power model, where sediment is assumed to be transported immediately out of the model domain. The second end-member model is the transport model and it takes the form of a diffusion equation, assuming that the sediment flux is a function of the water flux and slope. We find that both of these end-member models have a response time that has a proportionality to the precipitation rate that follows a negative power law. However, for the stream power model the exponent on the water flux term must be less than one, and for the transport model the exponent must be greater than one, in order to match the observed concavity of natural systems. This difference in exponent means that the transport model generally responds more rapidly to an increase in precipitation rates, on the order of $10^5$ years for post-perturbation sediment fluxes to return to within 50 % of their initial values, for theoretical landscapes with a scale of $100 \times 100$ km. Additionally from the same starting conditions, the amplitude of the sediment flux perturbation in the transport model is greater, with much larger sensitivity to catchment size. An important finding is that both models respond more quickly to a wetting event than a drying event, and we argue this asymmetry in response time has significant implications for depositional stratigraphies. Finally, we evaluate the extent to which these constraints on response times and sediment fluxes from simple models help us understand the geological record of landscape response to rapid environmental changes in the past, such as the Paleocene-Eocene thermal maximum (PETM). In the Spanish Pyrenees, for instance, a relatively rapid, 10 to 50 kyr, duration of deposition of gravel is observed for a climatic shift that is thought to be towards increased precipitation rates. We suggest the rapid response observed is more easily explained through a diffusive transport model because, (1) the model has a faster response time, consistent with the documented stratigraphic data; (2) there is a high-amplitude spike in sediment flux; and (3) the assumption of instantaneous transport is difficult to justify for the transport of large grain sizes as an alluvial bed-load. Consequently, while these end-member models do not reproduce all the complexity of processes seen in

real landscapes, we argue that variations in long-term erosional dynamics within source catchments can fundamentally control when, how and where sedimentary archives can record past environmental change.

## 1 Introduction

How river networks form and how landscapes erode remains a basic research question despite more than a century of experimentation and study. At a fundamental level, the root of the problem is a lack of an equation of motion for erosion derived from first principles (e.g Dodds and Rothman, 2000). A range of heuristic erosion equations have however been proposed, from stochastic models (e.g. Banavar et al., 1997; Pastor-Satorras and Rothman, 1998) to deterministic models based on the St. Venant shallow water equations (e.g. Smith and Bretherton, 1972; Izumi and Parker, 1995; Smith, 2010), diffusive *transport-limited* conditions (e.g. Whipple and Tucker, 2002), or the stream power law (e.g. Howard and Kerby, 1983; Whipple and Tucker, 2002; Willett et al., 2014, among many others). These models, in various forms, have been explored to try to understand how landscape evolves and responds to tectono-environmental change. In general terms, numerical studies have found that landscape typically recover from a shift in tectonic uplift after $10^5$ to $10^6$ years (reviewed in Romans et al., 2016). These apparently long response timescales to tectonic perturbations have been supported by field observations of landscapes upstream of active faults (e.g. Whittaker et al., 2007; Cowie et al., 2008; Whittaker and Boulton, 2012), although the precise appropriateness of any time-integrated erosion law to specific field sites is not always easy to establish. Sediment flux response times for the advective stream power law have been previously characterised by Whipple (2001) and Baldwin et al. (2003), and for the transport model they have been studied by Armitage et al. (2011) and Armitage et al. (2013), but not systematically or using 2-D models. Furthermore, to our knowledge no comparison between the transport model has been previously made.

The response of landscapes and sediment routing systems to a change in the magnitude or timescale of precipitation rates is expected to depend on the long term erosion law implemented (Castelltort and Van Den Dreissche, 2003; Armitage et al., 2011, 2013). Some numerical modelling studies, based on treating erosion as a length dependent diffusive problem, suggest that landscape responses to a change in rainfall are also on the order of $10^5$ to $10^6$ years, similar to tectonic perturbations, although they produce diagnostically different stratigraphic signatures from the latter (e.g. Armitage et al., 2011). However, other modelling contributions with different assumptions suggest that response times to a precipitation change may be more rapid (Simpson and Castelltort, 2012), although field data sets remain equivocal (see Demoulin et al., 2017 for a recent review). In laboratory studies, a series of experiments where granular piles of a length scale of order of centimeters are eroded due to surface water, have demonstrated that a change in precipitation rate leads a period of adjustment of the landscape topography until a new steady-state is achieved (e.g. Bonnet and Crave, 2003; Rohais et al., 2011). These experiments use a mixture of granular silica of a mean diameter in between 10 and $20\,\mu\mathrm{m}$, that is eroded by water released from a fine sprinkler system above. Given the complexity of these experiments, unfortunately there have been insufficient different precipitation rates studied to fully understand how the recovery time-scale varies as a function of precipitation or other parameters.

It has been increasingly recognised over the last two decades that many basic geomorphic measures of catchments, such as the scaling between channel slopes and catchment drainage areas, are typically unable to distinguish the erosional processes behind their formation (e.g. Tinkler and Whol, 1998; Dodds and Rothman, 2000; Tucker and Whipple, 2002; Whipple, 2004). Erosion and transport can be described by equations that encompass both advective and diffusive processes (e.g. Smith and Bretherton, 1972) and at topographic steady state, it is very well-established that fluvial erosion models based on either of these two end-members can produce very similar river longitudinal profiles (e.g. Tucker and Whipple, 2002; van der Beek and Bishop, 2003).

Non-uniqueness or equifinality is a common problem when comparing the morphology generated from landscape evolution models (e.g. Hancock et al., 2016). Consequently, we aim to explore if the sediment flux responses of fluvial systems to a precipitation perturbation may be diagnostically different for the two end-member deterministic models across a range of parameter space. This issue is pertinent because within sedimentary basins, a change in the erosional dynamics upstream could be recorded by changes in the total sediment volumes stored in sedimentary basins (e.g. Allen et al., 2013; Michael et al., 2014); in sediment delivery or sediment accumulation rates linked to landscape response times (Foreman et al., 2012; Armitage et al., 2015) and/or in the grain-sizes deposited as a function of sediment flux output (Paola et al., 1992; Armitage et al., 2011; Whittaker et al., 2011; D'Arcy et al., 2016).

In this article we make a comparative study between the transport and stream power model to further explore the potential differences between these two end-member hypothetical landscape evolution models. We will focus on the transient period of adjustment to a perturbation in precipitation rates, and using end-member numerical models attempt to evaluate how the response time varies as a function of the model forcing. To this end we aim to find the model parameters that generate similar landscape morphologies such that we can subsequently explore how the same end-member models respond to change in precipitation rate. We believe that the results of this study have implications for understanding the responses of landscapes to past change in climate, and could potentially be compared with and tested against further laboratory experiments.

## 2 Methods

### 2.1 Erosion within a single dimension system

We aim to understand the effects of the most basic assumptions of mass transport in landscape evolution on the sediment flux record. In other words, how do the response times vary for the advective stream power law and the diffusive transport model? To this end we derive the two models from first principles to demonstrate clearly how, from the same starting point, the fundamental assumptions made about mass transport initially give rise to very different model equations. We use this framework as a context for our investigation of an eroding system responding to precipitation change. We first define a one-dimensional system from which the basic equations can be assembled. Following Dietrich et al. (2003) we define a landscape of elevation $z$ composed of bedrock, thickness $\eta$ (units of m), and a surface layer of sediment with thickness $h$ (units of m; see Figure 1). This landscape is forced externally through uplift rate $U$ (units of $\mathrm{m\,yr^{-1}}$). The bedrock is transferred into sediment through erosion at a rate $E$ (units of $\mathrm{m\,yr^{-1}}$) and the sediment is transported across the system with a flux $q_s$ (units of $\mathrm{m^2\,yr^{-1}}$).

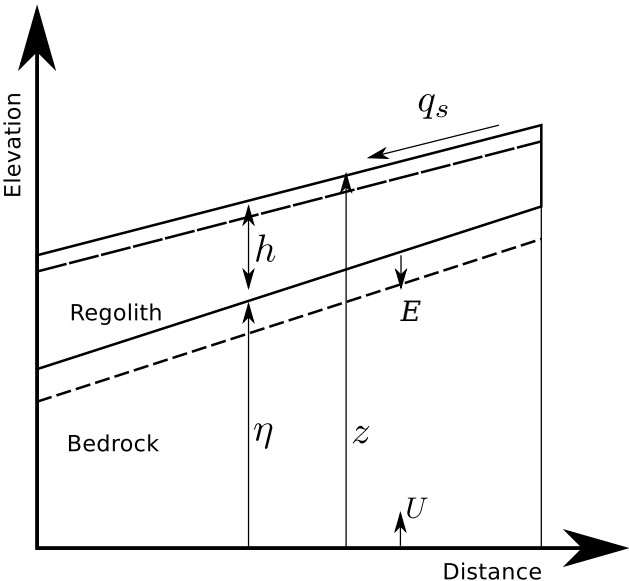

**Figure 1.** Diagram showing the conservation of mass within a 2-D domain, where mass enters the system through uplift, $U$ (units of $\mathrm{m\,s^{-1}}$), and exists as sediment transported, $q_s$ ($\mathrm{m^2\,s^{-1}}$) out of the domain. $E$ ($\mathrm{m\,s^{-1}}$) is the rate of production of regolith, $h$ (m) is the thickness of regolith, $\eta$ (m) is the bedrock elevation and $z$ (m) is the total elevation.

Assuming that the density of sediment and bedrock are equal, then the change in bedrock thickness is,

$$\partial_t \eta = U - E, \tag{1}$$

and the rate of change in sediment thickness is,

$$\partial_t h = E - \partial_x q_s. \tag{2}$$

5 It then follows that the rate of change in landscape elevation is,

$$\partial_t z = \partial_t \eta + \partial_t h. \tag{3}$$

It is important to realise that to solve Equation 3, we are required to make some assumptions that fundamentally affect the erosional dynamics of the modelled system, and we illustrate this below.

One basic assumption to make is that there is always a supply of transportable sediment, then we can follow through with
10 the summation in equation 3 giving,

$$\partial_t z = U - \partial_x q_s. \tag{4}$$

This may be appropriate when modelling the transport of sediment along the river bed and when considering the formation of alluvial fans (e.g. Paola et al., 1992; Whipple and Tucker, 2002; Guerit et al., 2014). In the absence of surface water we can

assume that sediment flux is simply a function of local slope $q_s = -\kappa \partial_x z$, where $\kappa$ is the hill slope diffusion coefficient. In the presence of flowing water then the sediment flux is a function of the flowing water and local slope $q_s = -cq_w^\delta (\partial_x z)^\gamma$ where $c$ is the transport coefficient (units $(\text{m}^2 \text{ yr}^{-1})^{1-n}$), $q_w$ is the water flux per unit width (units $\text{m}^2 \text{ yr}^{-1}$) and the exponents $\delta > 1$ and $\gamma \geq 1$ are dependent on how sediment grains are transported along the bed (Smith and Bretherton, 1972; Paola et al., 1992).

Furthermore, $\delta > 1$ is required to create concentrated flow (Smith and Bretherton, 1972). The change in landscape elevation is then given by,

$$\partial_t z = U + \partial_x \left( \kappa \partial_x z + cq_w^\delta (\partial_x z)^\gamma \right). \tag{5}$$

which can be written as,

$$\partial_t z = U + \partial_x \left( \left[ \kappa + cq_w^\delta (\partial_x z)^{\gamma-1} \right] \partial_x z \right). \tag{6}$$

Equation 6 is non-linear in the case that $\gamma \neq 1$. In deriving this equation of elevation change due to sediment transport we have simply summed the two terms for sediment flux, the linear and potentially non-linear slope dependent terms. This summation has been done as it is the simplest way to generate landscape profiles that have the desired convex and concave elements observed in natural landscapes (Smith and Bretherton, 1972).

To solve this equation in one dimension we assume that the water flux is a function of the precipitation transported down the

river network. The water collected is taken from the upstream drainage area, $a$, which is related to the main stream length, $l$, by $l \propto a^h$ where $h$ is the exponent taken from the empirical Hack's law (Hack, 1957). The main stream length is related to the longitudinal length of the catchment by, $l \propto x^d$ where $1 \leq d \leq 1.1$ (Tarboton et al., 1990; Maritan et al., 1996). Therefore, we can write that $x \propto a^{h/d}$, and the water flux is the precipitation rate, $\alpha$ units ($\text{m yr}^{-1}$), multiplied by the length of the drainage system,

$$q_w = k_w \alpha x^p \tag{7}$$

where $k_w$ is a constant of proportionality with units $\text{m}^{1-p}$, and $p = d/h$. Furthermore it is observed that river catchments are typically longer than they are wide, and so $p < 2$ (Dodds and Rothman, 2000). Therefore given that $0.5 < h < 0.7$ (e.g. Rigon et al., 1996) then $1.4 < p < 2$, and the transport model (equation 6) becomes,

$$\partial_t z = U + \partial_x \left( \left[ \kappa + ck_w \alpha^\delta x^{p\delta} (\partial_x z)^{\gamma-1} \right] \partial_x z \right). \tag{8}$$

For simplicity, we will also assume $k_w = 1$ and vary $c$ when exploring the model behaviour.

However, returning to equation 3, it is clear that the transport model is not the only solution. If we assume that rate of change in sediment thickness is zero over geological time scales, which is to say all sediment created is transported out of the model domain, than Equation 3 becomes,

$$\partial_t z = U - E. \tag{9}$$

This assumption has been made previously when studying small mountain catchments, where the river may erode directly into the bed-rock. However, recent numerical studies, such as Rudge et al. (2015), have expanded this model to cover continent-scale landscapes.

It is clearly plausible to suppose that erosion is primarily due to flowing water, so the assumption of geologically instantaneous transport may well be valid for mass that is transported as suspended load within the water column. Such an assumption is less clear for bed-load transport. We can assume that the speed at which suspended loads travel down system is a function of the height achieved within each hop, which is a function of the water depth, settling velocity and flow velocity. For small grains, $< 1\,\mathrm{mm}$, the settling velocity is given by the force balance between the weight of the grain and the viscous drag given by Stokes law (Dietrich, 1982). For a particle of diameter $1 \times 10^{-4}\,\mathrm{m}$ and density $2800\,\mathrm{kg\,m^{-3}}$ the settling velocity is $\sim 0.01\,\mathrm{m\,s^{-1}}$. Therefore the distance traveled assuming a flow velocity of $1\,\mathrm{km\,hr^{-1}}$ and an elevation of suspension of $1\,\mathrm{m}$ is roughly $3\,\mathrm{km}$. Using a similar argument the travel distance of a sediment grain typical of the Bengal Fan is estimated to be $\sim 10^4\,\mathrm{m}$ (Ganti et al., 2014). This suggests that rapid transport of sediment across a continent is possible.

The percentage of mass transported in suspension may also be quite significant. For a small Alpine braided river it was found that the majority of mass was transported as suspended load (Meunier et al., 2006), and for the river systems draining the Tian Shan, China, $70\,\%$ of mass is transported as suspended and dissolved load (Liu et al., 2011). Therefore significant mass may be transported rapidly, geologically instantaneously, down system suggesting that the assumption that $\partial_t h \sim 0$ may be valid in some circumstances.

Assuming surface flow is the primary driver of landscape erosion and that positive $x$ is in the downstream direction then erosion, $E$, as a function of the power of the flow to detach particles of rock per unit width can be written as,

$$E = -k_b \rho_w g q_w^m \left( \partial_x z \right)^n, \tag{10}$$

where $k_b$ is a dimensional constant that parameterises bedrock erodability (Howard and Kerby, 1983; units $(\mathrm{m^2\,yr^{-1}})^{1-m}\,\mathrm{yr\,kg^{-1}}$), $\rho_w$ is water density, $g$ is gravity, and $m$ and $n$ are constants. The exponent $m \sim 0.5$, as it is a function of how the stream flow width is proportional to the water flux (e.g. Lacey, 1930; Leopold and Maddock, 1953; Whittaker et al., 2007). The exponent $n > 0$ acts upon the slope.

In two dimensions the change in elevation is then given by,

$$\partial_t z = U + k q_w^m \left( \partial_x z \right)^n, \tag{11}$$

where the constant $k$ lumps together the other constants (units $\mathrm{m^{-1}\,(m^2\,yr^{-1})^{1-m}}$), and if $n \neq 1$ equation 11 becomes non-linear. Using a version of equation 11 to invert river profiles for uplift histories, it is argued by some authors that $n$ is close to unity (Rudge et al., 2015). However, certain river profiles may arguably be indicative of $n > 1$ (Lague, 2014). Furthermore if $n > 1$, equation 11 becomes non-linear and the model response to precipitation rate change will become a function of both uplift and precipitation rates (Whipple, 2001).

To solve equation 11 in 1D, as before we will assume that $q_w = k_w \alpha x^p$ where $1.4 < p < 2$. The stream power law for landscape erosion in 1-D is then,

$$\partial_t z = U + k_p \alpha^m x^{mp} \left( \partial_x z \right)^n, \tag{12}$$

where $k_p = k k_w \rho_w g$ (units $\mathrm{m^{-p}\,(m^2\,yr^{-1})^{1-m}}$).

We have demonstrated two different fundamental equations for change in elevation in 2-D (equations 6 and 11) and the equivalent 1-D forms (equations 8 and 12). These two models of elevation change differ in that equation 11 is an advection equation and equation 6 is a diffusion equation. This means that the time evolution of equation 11 would be a migrating wave of erosion traveling either up or down the catchment (Braun et al., 2015). This wave could also potentially take the form of a shock-wave, where due to the change in gradient, the lower reaches of the migrating wave could travel faster than the upper reaches, creating a breaking wave (Smith et al., 2000; Pritchard et al., 2009). The time evolution of equation 6 is very different because here the evolution is dominated by diffusive processes. The diffusion coefficient is a function of down-system collection of water, which can lead to the concentration of flow and the creation of realistic morphologies (Smith and Bretherton, 1972). It is not however completely established how the transport model responds differently to changes in precipitation forcing in comparison to the stream power model.

## 2.2 Linear and non-linear solutions

If $n = 1$ (equations 11 and 12) and $\gamma = 1$ (equations 6 and 8) then the models are linear, and we can solve the equations both analytically, and in 1-D and 2-D numerical schemes. For the stream power model we use an implicit finite difference scheme (Braun and Willett, 2013) and for the transport model we use an explicit finite element scheme with linear elements (Simpson and Schlunegger, 2003). If $n \neq 1$ and if $\gamma \neq 1$ the equations become non-linear. In this case the numerical solutions can become unstable for simple explicit schemes, and may suffer from too much numerical diffusion for implicit schemes, unless the size of the time step is limited by the appropriate Courant-Friedrichs-Lewy (CFL) condition (Campforts and Covers, 2015). Given the short time steps required to obtain an accurate solution, we explore the non-linear solutions for erosion down a river long profile in 1-D. We solve for the stream power model (equation 12) using an explicit total variation diminishing scheme with the appropriate CFL condition (Campforts and Covers, 2015). For the transport model (equation 8) we use an explicit finite element model with quadratic elements and the appropriate CFL condition to find a stable solution.

## 2.3 Generalizing to a two dimensional system

To solve equations 6 and 11 over a 2-D domain requires an algorithm to route surface flow down the landscape. In our case, to explore how a model landscape responds to change in precipitation rate we will make the simplest assumption available; that water flows down the steepest slope. We then solve for equation 11 using the numerical model Fastscape (Braun and Willett, 2013), with a resolution of 1000 by 1000 nodes for a 100 by 100 km domain, giving a spatial resolution of 100 m. Erosion by sediment transport in 2D is solved following the MATLAB model of Simpson and Schlunegger (2003), which is available from Simpson (2017). We solve Equation 6 on a triangular grid with a resolution of 316 by 316 nodes for a 100 by 100 km domain, giving a spatial resolution of the order of 300 m. We also explored how the models evolve for a domain that is 500 by 500 km in size. The time step used for both models is 10 kyrs.

We will explore how an idealized landscape evolves under uniform uplift at a rate of $0.1\,\mathrm{mm\,yr^{-1}}$. The initial condition is of a flat surface with a small amount of noise added to create a roughness. The boundary conditions are of fixed elevation at the left and right sides, and of no flow at the sides. To explore the response of the two models to change in precipitation rate

we start the model with an initial precipitation rate of $\alpha_0 = 1\,\mathrm{m\,yr^{-1}}$. For the linear models we then increase or decrease the precipitation rate to a new value, $\alpha_1$, after $10\,\mathrm{Myr}$ of model run time. This is to be sure that the steady state has been reached before applying the perturbation. For the non-linear models (Sections 3.3 and 3.4), the precipitation rate is changed after 5 Myr as in this case steady state was reached earlier. As the coefficients $c$ and $k$ have units that are related to the exponents $\delta$ and $m$ in equations 6 and 11 respectively (e.g. Whipple and Meade, 2006; Armitage et al., 2013), when modelling increasing values of $\delta$ and $m$ the coefficients are likewise increased.

The response time for the transport model scales by the effective diffusivity, and can be given by,

$$\tau_t = \frac{L^2}{\kappa + cq_w^\delta} \tag{13}$$

where $L$ is the model length scale (in this case the length of the domain). For the stream power model the response time is a function of the velocity at which the kinematic wave travels up the catchment (e.g. Whipple and Tucker, 1999; Whipple, 2001). The response time is therefore given by the time it takes for this wave of incision to travel up the catchment length, $l_c$,

$$\tau_{sp} = \frac{l_c}{kq_w^m} \tag{14}$$

Therefore we expect the response time to be a function of the choice of both the constants $c$ and $k$, and the exponents $\delta$ and $m$ within both models. The effect of varying the coefficients $m$ and $\delta$ independently has been previously explored (e.g. Whipple and Meade, 2006; Armitage et al., 2013), and we therefore will not do so in detail again here. Instead we aim to compare the two models, and therefore search for the values of $c$, $k$, $m$ and $\delta$ that generate similar topography at steady state. This steady state is then perturbed by a change in precipitation rate.

## 2.4 Generating similar landscapes

It has been previously demonstrated that both end-member models can generate convex-up long profiles (e.g. Kirkby, 1971; Smith and Bretherton, 1972; Smith et al., 2000; Whipple and Tucker, 2002; Crosby et al., 2007). From solving both equations 8 and 12 where $\gamma = 1$ and $n = 1$ we find that in range $1 < \delta \leq 1.5$ and $0.3 \leq m \leq 0.7$ the two end-member models are comparable (see Appendix A). Given the possible additional degree of freedom introduced if we also vary $\gamma$ and $n$, it is clear that river-long profiles are not a unique identifier of erosional processes. However, in order to compare how the end-member models respond to change in precipitation rate, it is preferable to perturb catchments of a similar morphology. We will subsequently explore how the models, in their linear and non-linear forms, respond to a change in precipitation rates within the Results section.

### 2.4.1 Erosion by Sediment Transport

Six models have been run without a change in precipitation to find the steady state topography. The models explored are first a set of three with varying $\delta$ and constant $c$, i.e.; $\delta = 1.1$, 1.3 and 1.5 with $c = 10^{-4}\,(\mathrm{m^2\,yr^{-1}})^{1-\delta}$ (Figure 2a), and a set of three where $\delta$ and $c$ co-vary, i.e.; $\delta = 1.1$ with $c = 10^{-2}\,(\mathrm{m^2\,yr^{-1}})^{1-\delta}$, $\delta = 1.3$ with $c = 10^{-3}\,(\mathrm{m^2\,yr^{-1}})^{1-\delta}$, and $\delta = 1.5$ with $c = 10^{-4}\,(\mathrm{m^2\,yr^{-1}})^{1-\delta}$ (Figure 2b). These values are chosen because they generate response times within the range of

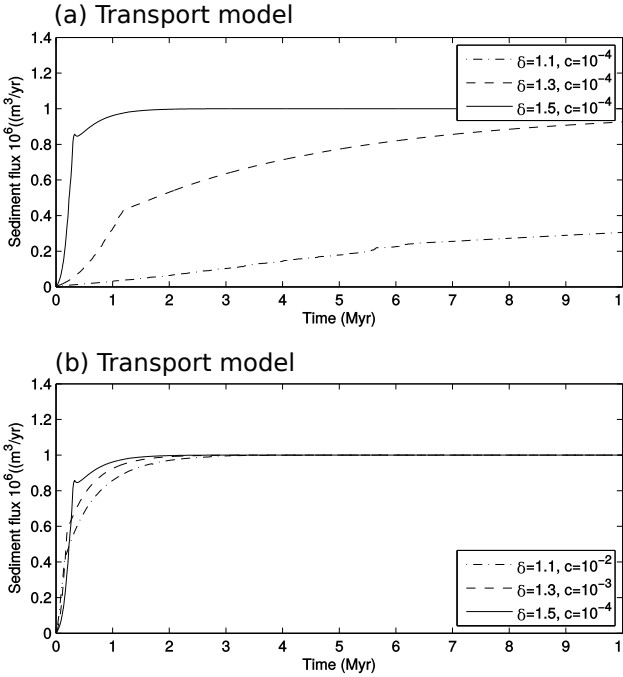

**Figure 2.** Sediment flux out of the model domain for the transport model for models where (a) $\delta = 1.1$, 1.3 and 1.5, $\kappa = 10^{-2}$ and $c = 10^{-4}\,(\mathrm{m}^2\,\mathrm{yr}^{-1})^{1-\delta}$, and (b) $\delta = 1.1$, 1.3 and 1.5, $\kappa = 10^{-2}$ and $c = 10^{-2}$, $10^{-3}$, and $10^{-4}\,(\mathrm{m}^2\,\mathrm{yr}^{-1})^{1-\delta}$.

observations from normal fault bounded sedimentary systems that have responded to changes in slip rate (Densmore et al., 2007; Armitage et al., 2011).

When the transport coefficient $c$ is the same for the three values of the exponent $\delta$ the model wind-up time increases with decreasing $\delta$, and takes several million years where $\delta < 1.5$ (Figure 2a). Steady state sediment flux is greater for increasing
$\delta$ when $c$ is kept constant. The dimensions (units) of $c$ depend on $\delta$ which means that the value of the coefficient $c$ must be adjusted when $\delta$ is changed to yield the same unit erosion rate per water flux, regardless of $\delta$ (see Armitage et al., 2013). Consequently, when $c$ is suitably adjusted the model can reach a steady state in a similar time for all three values of $\delta$ (Figure 2b).

We subsequently analyze the topography for the relationship between trunk river slope and drainage area, Figure 3, using
Topotoolbox2 (Schwanghart and Scherler, 2014). For the case where $\delta = 1.5$ the scaling between channel slopes and catchment drainage areas, the slope area exponent $\theta$, is equal to -0.42, and for $\delta = 1.3$, $\theta$ is equal to -0.23 (Figure 3b). The same value is calculated using the spatial transformation described within Perron and Royden (2012), commonly referred to as $\chi$-analysis (Table 1). Given the reduction in $\theta$ from $\delta = 1.5$ to 1.3, we did not analyze the case for $\delta = 1.1$ as the slope-area relationship will clearly lie below the observed range ($-0.7 < \theta < -0.35$; e.g. Snyder et al., 2000; Wobus et al., 2006). Therefore, for river
networks defined by routing water down the steepest slope of descent, the transport model can create catchment morphologies that have a concavity similar to that observed in nature if $\delta \sim 1.5$.

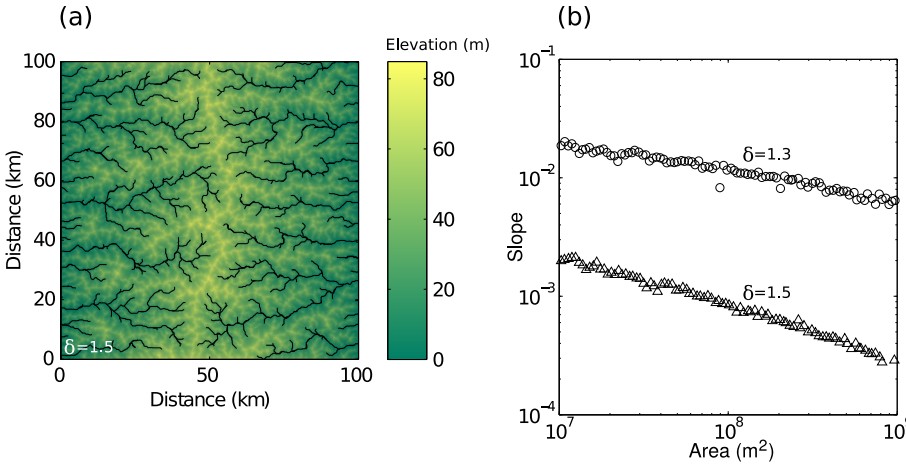

**Figure 3.** (a) Steady state topography, after $10\,\mathrm{Myr}$, for the transport model where $\delta = 1.5$ and $c = 10^{-4}\,(\mathrm{m^2\,yr^{-1}})^{1-\delta}$. (b) Slope area relationship for transport model for $\delta = 1.3$ and $\delta = 1.5$.

**Table 1.** Slope area relationship for trunk streams derived using $\chi$-analysis (Perron and Royden, 2012)

| sediment transport | $k_S$ | $\theta$ |
|---|---|---|
| $\delta = 1.3$ | 0.86 | -0.23 |
| $\delta = 1.5$ | 1.76 | -0.42 |
| stream power | | |
| $m = 0.3$ | 0.95 | -0.29 |
| $m = 0.5$ | 6.52 | -0.46 |
| $m = 0.7$ | 71.42 | -0.68 |

### 2.4.2 Comparison to Erosion by Stream Power

In order to provide a comparison for the morphology of the transport model we explore how the stream power model evolves to a steady state. The landscape derived from the stream power model, equation 11, evolves towards a steady state with a slightly different behaviour in comparison to the transport model (Figure 4). As before we run six models where in this case the first set of three are $m = 0.3$, 0.5 and 0.7 with $k = 10^{-5}\,\mathrm{m^{-1}\,(m^2\,yr^{-1})^{1-m}}$ (Figure 4a). The second set of three are of $m = 0.3$ with $k = 10^{-4}\,\mathrm{m^{-1}\,(m^2\,yr^{-1})^{1-m}}$, $m = 0.5$ with $k = 10^{-5}\,\mathrm{m^{-1}\,(m^2\,yr^{-1})^{1-m}}$, and $m = 0.7$ with $k = 10^{-6}\,\mathrm{m^{-1}\,(m^2\,yr^{-1})^{1-m}}$ (Figure 4b). This range of $m$ is chosen as it spans the range of observed concavities within catchments. As with the transport model the coefficient $k$ can be adjusted along with $m$ as they are related, where increasing $k$ reduces the model wind-up time (Figure 4b). Decreasing the exponent $m$ increases the timescale taken to reach a steady state (Figure 4a), however by varying $k$ by a factor of 100, the steady state sediment flux is reached within $3\,\mathrm{Myrs}$ for the three values of $m$ (Figure 4b).

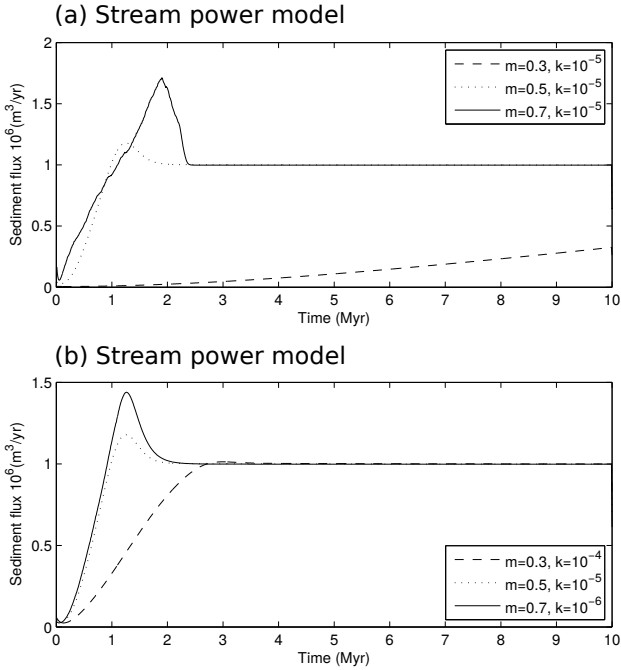

**Figure 4.** Sediment flux out of the model domain for the stream power model for models where (a) $m = 0.3$, 0.5 and 0.7, and $k = 10^{-5}\,(\mathrm{m^2\,yr^{-1}})^{1-m}$, and (b) $m = 0.3$, 0.5 and 0.7, and $k = 10^{-4}$, $10^{-5}$, and $10^{-6}\,\mathrm{m^{-1}}\,(\mathrm{m^2\,yr^{-1}})^{1-m}$.

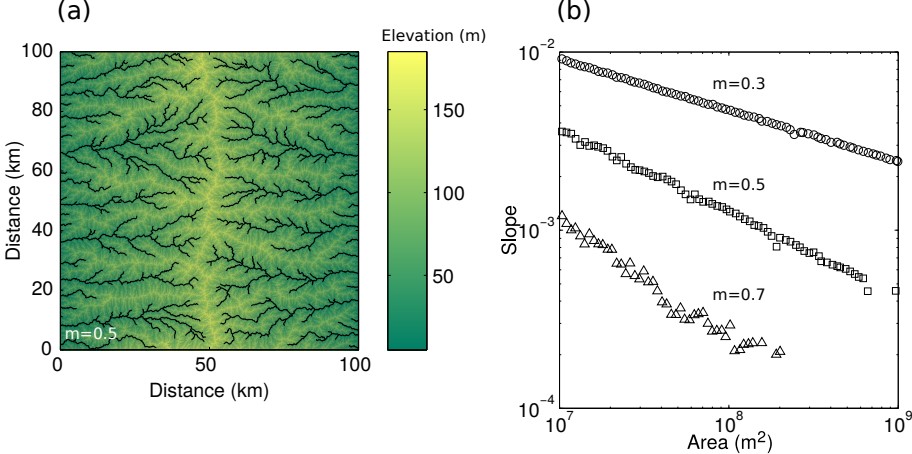

**Figure 5.** (a) Steady state topography, after 10 Myr, for the transport model where $m = 0.5$ and $k = 10^{-5}\,\mathrm{m^{-1}}\,(\mathrm{m^2\,yr^{-1}})^{1-m}$. (b) Slope area relationship for transport model for $m = 0.3$, 0.5 and 0.7.

Following the previous examples, we analyze the topography for the relationship between trunk river slope and drainage area (Figure 5). Both the transport model and the stream power model can create landscapes with similar slope-area relationships

calculated using the $\chi$-analysis approach (Table 1). For both models, the value of the intercept, $k_s$, and the gradient, $\theta$, are of similar magnitudes for $\delta = 1.5$ and $m = 0.5$. Absolute elevation for the model shown in Figure 5a is higher than the transport model example due to the larger value of $k$ relative to $c$. However, importantly, both models can create similar landscape morphologies at steady state.

## 3 Results

The stream power and transport model can both fit observed slope-area relationships of the present day landscape morphology, e.g. $\theta$ ranging from -0.35 to -0.7 (Snyder et al., 2000; Wobus et al., 2006), when the water flux exponent is $m \sim 0.5$ or $\delta \sim 1.5$ for the stream power and transport model respectively. Therefore, both models may be a reasonable representation of how, on a gross scale, a landscape erodes. We therefore keep the exponents in the range $0.3 \leq m \leq 0.7$ and $1.3 \leq \delta \leq 1.5$, and explore how the models, in their linear and non-linear forms, respond to a change in precipitation rates.

### 3.1 Response to precipitation rate reduction

When the model is perturbed by a change in precipitation rate the sediment flux output will first change, as the erosive power changes (e.g. Figure 6). The model will subsequently return to the steady-state output, as the slope of the fluvial system will adjust to the new precipitation rate, and the landscape will re-achieve the same steady-state. In Figure 6a we display the response of erosion for the transport model, in terms of sediment flux out of the model domain, for a reduction in precipitation rate from 1 to $0.5\,\mathrm{m\,yr^{-1}}$ at $10\,\mathrm{Myr}$ of model evolution. We explore how the transport model responds for $\delta = 1.5$, $c = 10^{-4}$; $\delta = 1.3$, $c = 10^{-3}$ as these two values of $\delta$ generate reasonable slope area relationships (Figure 3b, Table 1). The response to a reduction in precipitation is similar for the two model parameter sets, with the flux initially reducing by a half and then recovering to within $10\,\%$ of steady state values within $\sim 2\,\mathrm{Myr}$ (Figure 6a; see Table 2). Changing the transport coefficient, $c$, does not affect predicted gradient of catchment slope versus catchment area (see Appendix A, Figure 16). However, changing $c$ changes the model elevation (Figure 16). Furthermore, the larger the value of $c$ the faster the response (Equation 13; see Armitage et al., 2013). A small increase in the exponent $\delta$ will strongly reduce response times, as it will increase the water flux term (Equation 13). Therefore an order of magnitude decrease in $c$ counters the change in $\delta$ for the two model sets (Figure 6a). For the values chosen both models respond at a similar rate to the change in precipitation (Figure 6a; see Table 2).

The response of the stream power model to an identical reduction in precipitation at a model time of $10\,\mathrm{Myr}$ takes a similar form, with an initial decrease in sediment flux out followed by a gradual recovery (Figure 6b). In a similar manner as the transport model, response is a function of the exponent $m$ and the coefficient $k$ (Equation 14). We have modeled three parameter sets: $m = 0.3$ and $k = 10^{-4}$, $m = 0.5$ and $k = 10^{-5}$, and $m = 0.7$ and $k = 10^{-6}$ (Figure 6b). The response time to achieve return to $10\,\%$ of the steady state sediment flux varies from $3\,\mathrm{Myr}$ in the case of $m = 0.3$ to less than $1\,\mathrm{Myr}$ when $m = 0.7$. As well as response time being longer for smaller values of $m$, the peak magnitude of the flux response is reduced for smaller values of $m$ (Figure 6b).

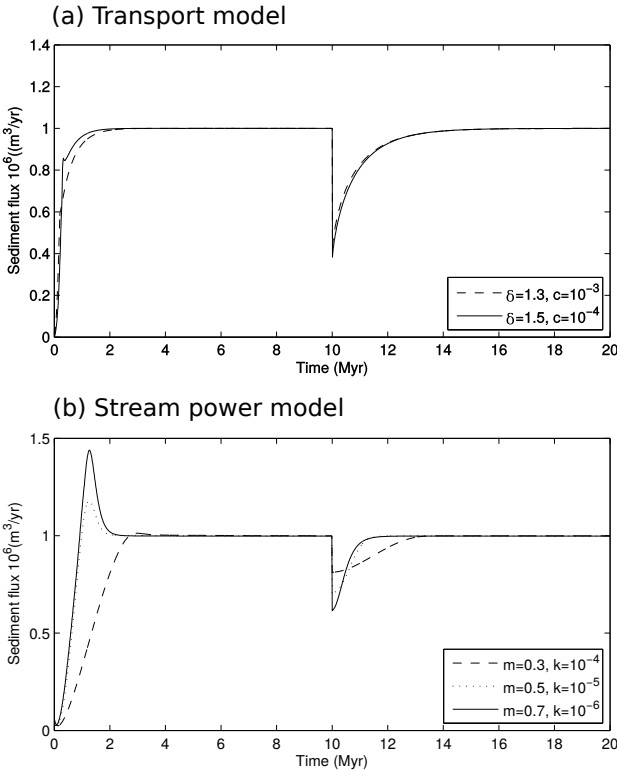

**Figure 6.** Response of the transport and stream power model to a reduction in precipitation rate (a) Sediment flux for the transport model for a step reduction in precipitation from 1 to $0.5\,\mathrm{m\,yr^{-1}}$ after 10 Myr. Two models are plotted, where $\delta = 1.3$ and 1.5, $\kappa = 10^{-2}$ and $c = 10^{-3}$, and $10^{-4}\,(\mathrm{m^2\,yr^{-1}})^{1-\delta}$. (b) Sediment flux for the stream power model for a step reduction in precipitation from $\alpha_0 = 1\,\mathrm{m\,yr^{-1}}$ to $\alpha_1 = 0.5\,\mathrm{m\,yr^{-1}}$ after 10 Myr. Three models are plotted, where $m = 0.3, 0.5$ and 0.7, and $k = 10^{-4}, 10^{-5}$, and $10^{-6}\,\mathrm{m^{-1}}\,(\mathrm{m^2\,yr^{-1}})^{1-m}$

The magnitude of the response for all the runs is greater for the transport model when compared to the stream power model (Figure 6). Consequently, response time, while being a function of the transport coefficients $c$ and $k$ respectively, may still systematically differ between the two models: The transport model with $\delta = 1.5$ and $c = 10^{-4}$ generates a maximum model elevation of $\sim 240\,\mathrm{m}$, and the stream power model with $m = 0.5$ and $k = 10^{-5}$ generates a maximum elevation of $\sim 180\,\mathrm{m}$. These two models have a similar slope area relationship at steady state (Table 1) and are therefore comparable suggesting a faster response to a reduction in precipitation rates for the stream power model (Figure 6).

To explore how the difference in response time and magnitude is expressed in the landscape, we extract the river profiles of the main trunk systems for models where $\delta = 1.5$ and $m = 0.5$ during the responce to the reduction in precipitation rate while uplift rate is constant (Figures 7 and 8). For the transport model in which $\delta = 1.5$ and $c = 10^{-4}$, the catchment elevation increases to a new steady state that has an elevation that is roughly 2.6 times higher than the steady state elevation after 10 Myr (Figure 7). Just under half of this new topographic elevation is achieved within the first 500 kyr. In contrast, for the stream power model where $m = 0.5$ and $k = 10^{-5}$, the steady state topography is achieved within a fraction of the time when

**Table 2.** Response to change in precipitation rate where $\alpha_1$ represents the value the precipitation rate changes to from $\alpha_0 = 1\,\mathrm{mm\,yr}^{-1}$. Response time is given for two model sizes, 100 and 500 km, and as the time for the model to recover by half and a tenth towards the steady state sediment flux.

| $L = 100\,\mathrm{km}$ | transport | | detachment | |
|---|---|---|---|---|
| $\alpha_1$ | $\tau_{1/2}$ | $\tau_{1/10}$ | $\tau_{1/2}$ | $\tau_{1/10}$ |
| $\mathrm{mm\,yr}^{-1}$ | Myr | Myr | Myr | Myr |
| 0.25 | 1.42 | 6.07 | 0.98 | 1.66 |
| 0.50 | 0.53 | 2.19 | 0.70 | 1.18 |
| 0.75 | 0.30 | 1.21 | 0.57 | 0.98 |
| 2.00 | 0.09 | 0.31 | 0.34 | 0.60 |
| $L = 500\,\mathrm{km}$ | transport | | detachment | |
| $\alpha_1$ | $\tau_{1/2}$ | $\tau_{1/10}$ | $\tau_{1/2}$ | $\tau_{1/10}$ |
| $\mathrm{mm\,yr}^{-1}$ | Myr | Myr | Myr | Myr |
| 2.00 | 0.17 | 0.64 | 0.34 | 0.60 |

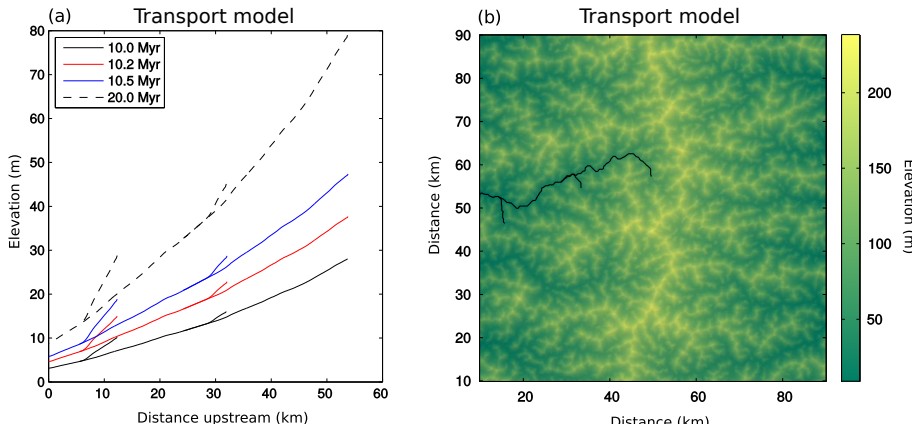

**Figure 7.** Transport model evolution due to a reduction in precipitation. (a) Selected river long profile response to change in precipitation. Black line is the profile just before a factor of two reduction in precipitation. The red and blue lines are 200 kyr and 500 kyr after the reduction in precipitation. The dashed black line is the steady state profile. (b) Trunk stream used for the analysis with the steady state elevation.

compared to the transport model. This is in line with the more rapid response of this model to a relative drying of the climate using these parameters (compare Figure 6a and b). Furthermore the increase in elevation due to the reduced surface water flux is only a factor of $\sim 1.2$, which is less than half of the increase for the transport model. Our results confirm that two different end-member erosion models, encompassing advective and diffusive phenomena, can produce landscapes with similar morphologies, if particular parameter sets are selected accordingly.

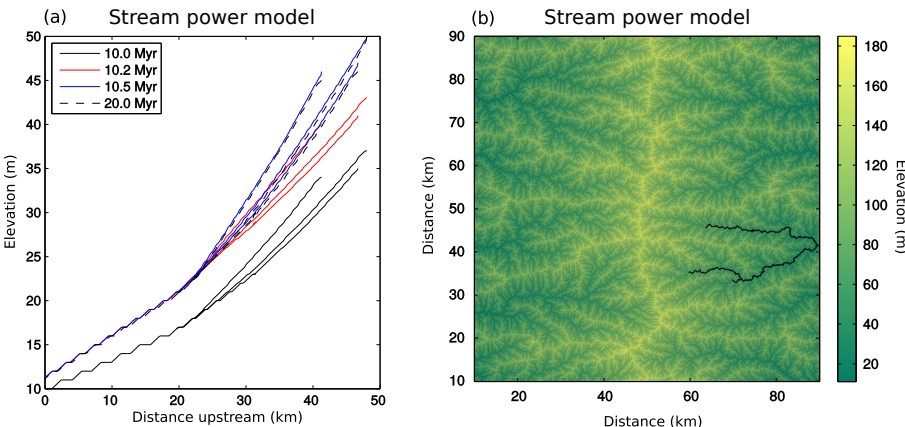

**Figure 8.** Stream power model model evolution due to a reduction in precipitation. (a) Selected river long profile response to change in precipitation. Black line is the profile just before a factor of two reduction in precipitation. The red and blue lines are 200 kyr and 500 kyr after the reduction in precipitation. The dashed black line is the steady state profile. (b) Trunk stream used for the analysis with the steady state elevation.

## 3.2 Response to different magnitudes of precipitation rate change

The response time of the transport model is known to be a function of the transport coefficient and the magnitude of the precipitation rate (c.f. Armitage et al., 2013). This behavior is displayed in Figure 9a, where the response of the transport model with $\delta = 1.5$ and $c = 10^{-4}$ for a change in precipitation from 1 to 0.25, 0.5, 0.75 and $2\,\mathrm{m\,yr}^{-1}$ is plotted. The response time, measured as the time for the sediment flux to recover by half and by 90 % to the steady state value, is shown additionally in Figure 10 as black solid and dashed lines respectively, and in Table 2. For a reduction to $0.25\,\mathrm{m\,yr}^{-1}$ the prediction is for a long response time of 6.07 Myr, while for an increase to $2\,\mathrm{m\,yr}^{-1}$ the prediction is a for rapid response time of 310 kyr for 90 % recovery towards previous sediment flux values. The equivalent half life, recovery by 50 % towards previous sediment flux values, is 1.42 Myr and 90 kyr.

The stream power model likewise has a response time that is a function of precipitation rate (Figure 9b). For a reduction to $0.25\,\mathrm{m\,yr}^{-1}$ the prediction is for a response time of 1.66 Myr, while for an increase to $2\,\mathrm{m\,yr}^{-1}$ the prediction is for recovery time of 600 kyr for 90 % recovery (Table 2). The equivalent half life is 0.98 Myr and 340 kyr (Table 2). The stream power model is therefore faster to recover for a reduction in precipitation rate yet slower to respond to an increase in precipitation rate. This is because the response time of the stream power model is more weakly a function of precipitation rate. Importantly, these results therefore suggest there is a fundamental asymmetry in the response timescale to a climate perturbation. The models suggest that it takes longer for surface processes to recover from drying event compared to a wetting event.

Both models display a response time that is a function of the precipitation rate (Figures 9 and 10). The relationship between precipitation rate and the transport model response can be expressed as,

$$\tau_t \propto \alpha^{-\delta} \tag{15}$$

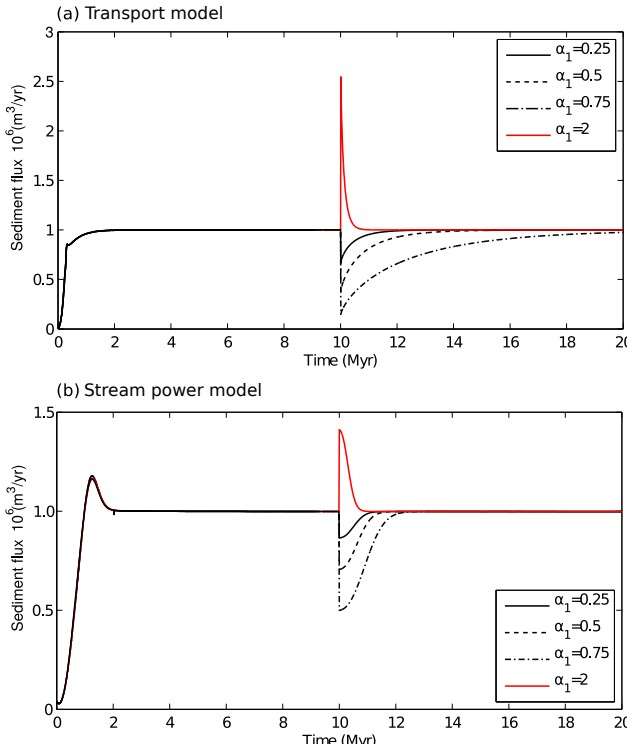

**Figure 9.** (a) Response of the transport model to change in precipitation rate. Equation 6 is solved for $\delta = 1.5$ and $c = 1 \times 10^{-4} \, (\mathrm{m^2 \, yr^{-1}})^{1-\delta}$. The precipitation rate is initially $\alpha_0 = 1 \, \mathrm{m \, yr^{-1}}$ and changes to $\alpha_1 = 0.25$, 0.5, 0.75 or $2 \, \mathrm{m \, yr^{-1}}$ after 10 Myr. (b) Response of the stream power model to change in precipitation rate. Equation 6 is solved for $m = 0.5$ and $k = 1 \times 10^{-5} \, \mathrm{m^{-1}} \, (\mathrm{m^2 \, yr^{-1}})^{1-m}$. The precipitation rate is initially $\alpha_0 = 1 \, \mathrm{m \, yr^{-1}}$ and changes to $\alpha_1 = 0.25$, 0.5, 0.75 or $2 \, \mathrm{m \, yr^{-1}}$ after 10 Myr.

where in this case $\delta = 1.5$. This proportionality is in agreement with our numerical model results, where the slope of trend for the transport model in the log-log plot is -1.5 (Figure 10).

In contrast, the response time of the stream power model is not as strongly inversely dependent on the precipitation rate (Figure 10). For this model, the response time is a function of the velocity at which the wave of incision travels up-stream. This velocity is directly related to the inverse of the water flux, $q_w^m$, which is in turn again a function of the drainage length and precipitation rate, $\alpha$. Therefore for the stream power model we can write that response time is,

$$\tau_{sp} \propto \alpha^{-m}. \tag{16}$$

This proportionality, which is in agreement with the approximate analytical solutions of Whipple (2001), is likewise in agreement with our numerical model results, where the slope of trend for the stream power model in the log-log plot is -0.5 (Figure 10). Consequently, for these two models, which were derived from the same starting point (Figure 1), and applied to catchments of similar topography and morphology, we find that above a certain magnitude of precipitation rate change, the transport model responds more rapidly than the stream power model and vice versa.

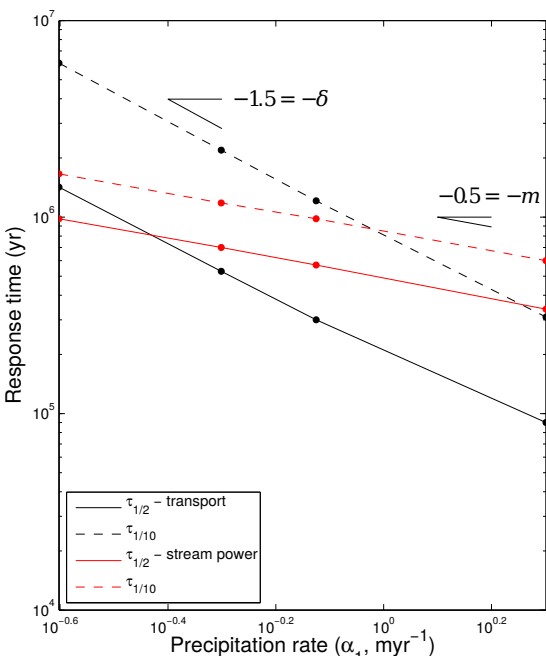

**Figure 10.** Log-log plot of response time to change to a precipitation rate $\alpha_1$ from an initial value of $\alpha_0 = 1\,\mathrm{m\,yr}^{-1}$ when the model domain is 100 by 100 km (see Table 2). $\tau_{1/2}$ is the time for the sediment flux to recover by a half of the magnitude change in sediment flux and $\tau_{1/10}$ is the time for the sediment flux to recover by 90 %.

The position of the critical point where the stream power model responds more rapidly than the transport model is a function of the water flux and the collection of coefficients. In the model comparison developed here, we have compared two model catchments that have similar slope area exponent, $\theta$ between -0.4 and -0.5 ($\delta = 1.5$ and $m = 0.5$) and model domain length of $L = 100\,\mathrm{km}$ giving catchments of roughly 50 km length. In this case the 90 % recovery of the sediment flux signal is predicted

5   to be more rapid for the transport model when compared to the stream power model for an increase in precipitation rate (Figure 10). If however the model domain is increased to $L = 500\,\mathrm{km}$ then it takes twice as long for the transport model to recover from an increase in precipitation rate from 1 to $2\,\mathrm{m\,yr}^{-1}$: 0.63 Myr compared to 0.31 Myr for $L = 100\,\mathrm{km}$ (Figure 11a and Table 2).

The stream power model is insensitive to the size of the model domain because of the particular choice of $m = 0.5$ and the shape of drainage network that forms under the assumptions of routing water down the steepest slope of descent (Figure

10   11b). Taking the drainage length to be directly proportional to the catchment area, $l_d \propto a$, and given that catchment length is proportional to drainage area raised to the Hack exponent, $h$, we can re-write equation 14 as,

$$\tau_{sp} \propto \frac{a^h}{(\alpha a)^m}. \tag{17}$$

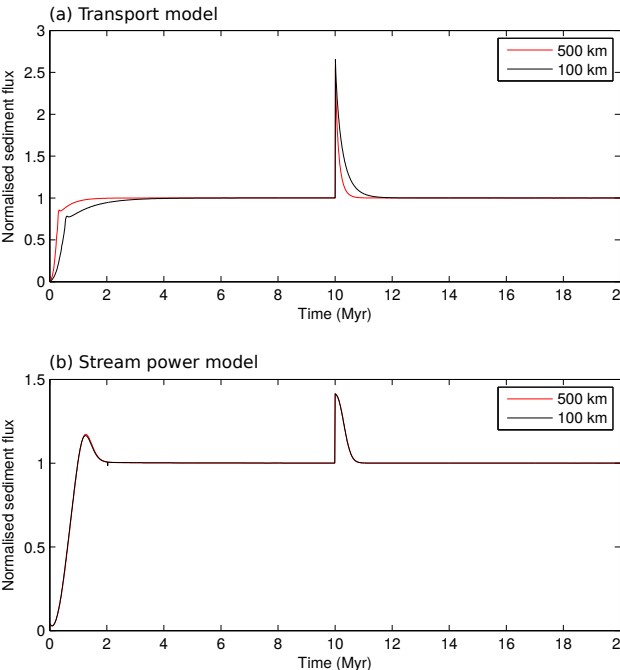

**Figure 11.** (a) Response of the transport model to change in precipitation rate for two different model dimensions, 100 by 100 km and 500 by 500 km. Equation 6 is solved for $\delta = 1.5$ and $c = 1 \times 10^{-4} \, (\mathrm{m^2 \, yr^{-1}})^{1-\delta}$. The precipitation rate is initially $\alpha_0 = 1 \, \mathrm{m \, yr^{-1}}$ and changes to $\alpha_1 = 2 \, \mathrm{m \, yr^{-1}}$ after 10 Myr. (b) Response of the stream power model to change in precipitation rate for two different model dimensions, 100 by 100 km and 500 by 500 km. Equation 6 is solved for $m = 0.5$ and $k = 1 \times 10^{-5} \, \mathrm{m^{-1}} \, (\mathrm{m^2 \, yr^{-1}})^{1-m}$. The precipitation rate is initially $\alpha_0 = 1 \, \mathrm{m \, yr^{-1}}$ and changes to $\alpha_1 = 2 \, \mathrm{m \, yr^{-1}}$ after 10 Myr.

Therefore, in the case that $h = 0.5$ and $m = 0.5$, as in the numerical model here, the response time becomes independent of system length (c.f. Whittaker and Boulton, 2012). If $h < m$ then response times would reduce with increasing drainage basin size, and if $h > m$ then response times would increase with drainage basin size. There is good empirical evidence for $0.5 < h < 0.7$ (e.g. Rigon et al., 1996), which fundamentally controls the plan view shape of catchments, yet there is not a

5 complete consensus on the value of $m$ (see Lague, 2014; Temme et al., 2017).

A final key difference between the transient sediment flux responses of the two models is that the peak magnitude of system response to a change in precipitation rate is systematically larger for the transport model (Figure 9). For an increase in precipitation rates from 1 to $2 \, \mathrm{m \, yr^{-1}}$, the sediment flux increases from $1 \times 10^6 \, \mathrm{m^3}$ to $2.5 \times 10^6 \, \mathrm{m^3}$ for erosion by sediment transport. This is three times greater than the equivalent increase for the stream power model. The reduction in sediment flux

10 is likewise larger for the transport model (Figure 9). Therefore, although response time is a function of precipitation rate, the magnitude of change is consistently larger for the transport model.

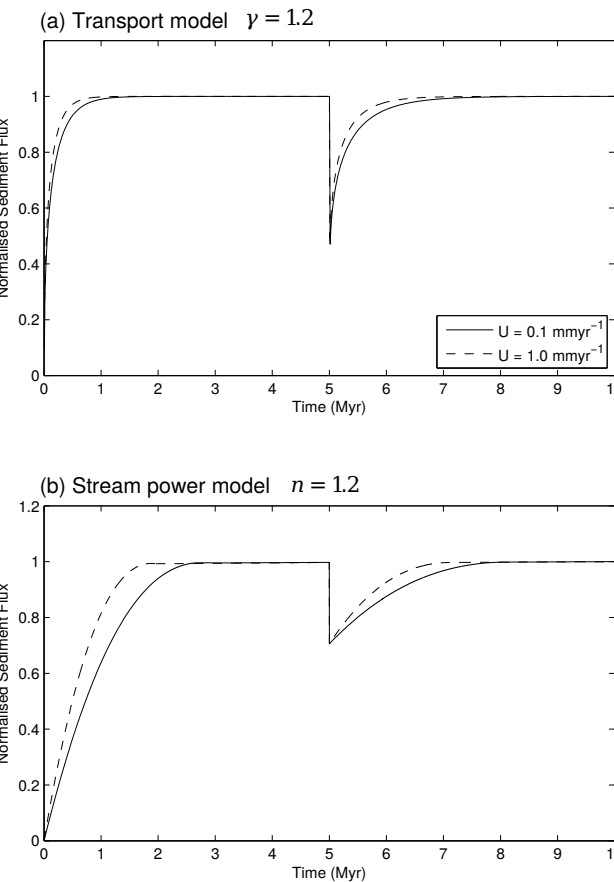

**Figure 12.** (a) Response of the transport model to change in precipitation rate for two values of uplift, 0.1 and $1.0 \, \mathrm{mm \, yr^{-1}}$. Equation 8 is solved for $\gamma = 1.2$, $\delta = 1.5$, $p = 1.1$ and $c = 5 \times 10^{-5} \, (\mathrm{m^2 \, yr^{-1}})^{1-\delta}$. The precipitation rate is initially $\alpha_0 = 1 \, \mathrm{m \, yr^{-1}}$ and changes to $\alpha_1 = 0.5$. (b) Response of the stream power model to change in precipitation rate for two values of uplift, 0.1 and $1.0 \, \mathrm{mm \, yr^{-1}}$. Equation 12 is solved for $n = 1.2$, $m = 0.5$, $p = 1.1$ and $k = 1 \times 10^{-4} \, \mathrm{m^{-1}} \, (\mathrm{m^2 \, yr^{-1}})^{1-m}$. The precipitation rate is initially $\alpha_0 = 1 \, \mathrm{m \, yr^{-1}}$ and changes to $\alpha_1 = 0.5$ after 5 Myr.

### 3.3 Non-linear response timescales

Up to this point we have compared how the models respond to a precipitation rate change when the solutions are linear. However, there is reasonable debate as to the value of the slope exponent $n$ in the stream power model (e.g. Lague, 2014; Croissant and Braun, 2014; Rudge et al., 2015) and likewise within the transport model it is plausible that the slope exponent

5    $\gamma > 1$. The response time for the stream power model for various values of $n$ has been explored within Baldwin et al. (2003). Here we expand on this by exploring the equivalent response times for the transport model. To explore the implications of the non-linearity introduced by relaxing the constraint that $n = 1$ and $\gamma = 1$ for both models, we solve equations 8 and 12 for

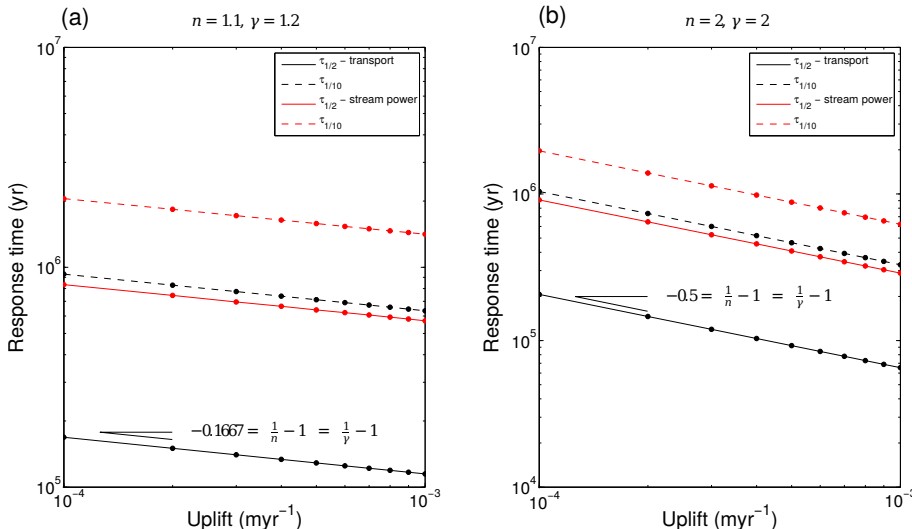

**Figure 13.** Log-log plot of response time for different slope exponents and uplift rates to change to a precipitation rate from an initial value of $\alpha_0 = 1\,\mathrm{m\,yr^{-1}}$ to $\alpha_1 = 0.5\,\mathrm{m\,yr^{-1}}$. $\tau_{1/2}$ is the time for the sediment flux to recover by a half of the magnitude change in sediment flux and $\tau_{1/10}$ is the time for the sediment flux to recover by 90 %. (a) Response time for the transport model (equation 8) and stream power model (equation 12) when the slope exponent $\gamma = 1.2$ and $n = 1.2$ respectfully. A linear trend is found with a gradient of -1.667. (b) Response time for the transport model and stream power model when the slope exponent $\gamma = 2$ and $n = 2$ respectfully. A linear trend is found with a gradient of -0.5.

$p = 1.1$, $\delta = 1.5$, $c = 5 \times 10^{-5}$, and $m = 0.5$, $k = 10^{-4}$, respectively with different uplift rates. We have modelled the response due to an uplift rate of between 0.1 and $1.0\,\mathrm{mm\,yr^{-1}}$ for the case where $\gamma = 1.2$ and $n = 1.2$ in equations 8 and 12 (Figure 12).

We find that for both the transport and stream power model, when the slope exponent is greater than one, the model response time is a function of uplift rate. The faster the rate of uplift, the faster the system responds to a change in precipitation rate. If the response time for a system recovery to steady state by 50 % or 10 % is plotted on a log-log plot against uplift rate we find that the response time is proportional to the uplift rate raised to a negative power (Figure 13). In the case of $n = 1.2$ or $\gamma = 1.2$ the slope of trend is -0.1667, and for $n = 2$ or $\gamma = 2$ the slope of trend is -0.5 (Figure 13). These slopes are in agreement with the approximate analytical solutions of Whipple (2001) and numerical models of Baldwin et al. (2003), i.e. the stream power response time $\tau_{sp}$ has a proportionality,

$$\tau_{sp} \propto U^{\frac{1}{n}-1} \tag{18}$$

and equivalently we infer from our numerical model (Figure 13) that the transport model response time as,

$$\tau_t \propto U^{\frac{1}{\gamma}-1}. \tag{19}$$

This implies that both models have the same form of response dependency on uplift rates. Therefore, regardless of the rate of uplift we should expect the transport model to respond more rapidly to a large increase in precipitation rate and the stream

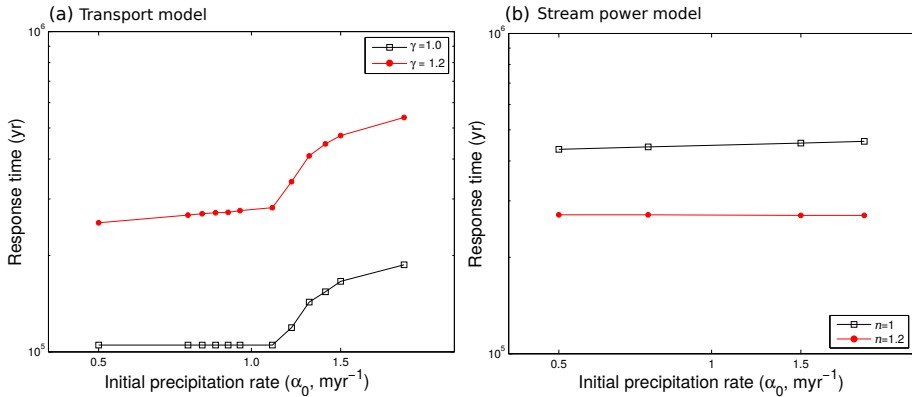

**Figure 14.** Log-log plots for the transport model and the stream power model in 1-D for a step change in precipitation rate, where the initial precipitation rate, $\alpha_0$ varies from 0.5 to $1.5\,\mathrm{m\,y}^{-1}$ and the final precipitation rate is fixed at $\alpha_1 = 1\,\mathrm{m\,y}^{-1}$. (a) Results for the transport model. (b) Results for the stream power model.

power model to respond more rapidly to a reduction in precipitation rate (Figure 10). Our results are also consistent with the field-based findings of Whittaker and Boulton (2012), who showed that landscape response times for rivers close to the detachment-limited end-member were shorter for terrain uplifted by faster slipping active faults.

### 3.4 Response time as a function of the initial precipitation rate

Up until this point we have only explored how the numerical models respond to a increase or decrease in precipitation rate by keeping the initial precipitation rate fixed at $\alpha_0 = 1\,\mathrm{m\,y}^{-1}$ and varying the final precipitation rate $\alpha_1$. In this final section we will now instead keep the final precipitation rate fixed at $\alpha_1 = 1\,\mathrm{m\,y}^{-1}$, and vary the initial precipitation rate $\alpha_0$ from values of 0.5 to $1.5\,\mathrm{m\,y}^{-1}$. We will focus again on the 1-D models, and look at the linear and non-linear cases with $n = 1.2$ and $\gamma = 1.2$.

   For the linear and non-linear transport model we find that if the initial precipitation is less than the final precipitation

($\alpha_0 < \alpha_1$) then the response time is not very sensitive to the initial precipitation rate (Figure 14a). If $\alpha_0 > \alpha_1$ then the response time is a function of the initial precipitation rate, but the relationship cannot be explained by a simple power law (Figure 14a). The change in response time as a function of the initial precipitation rate is however small compared to the change in response time as a function of the final precipitation rate.

   In the case of the linear and non-linear stream power model, the response time has a no dependence on the initial precipitation

rate, and is only a function of the final precipitation rate (Figure 14b). With all other parameters being held constant, the initial precipitation rate will set up the topography and hence slope of the pre-perturbation landscape. Elevations will be lower for higher precipitation rates, and the topographic gradient will be smaller. For the case of the stream power model, the change in erosion rates migrates up the catchment and so the old topography does not impact the response time. For the transport model, however, the remnant topography does have a small effect on the response time, but only if the previous precipitation rate was

higher than the new post perturbation precipitation rate.

## 4 Discussion

In deriving the two end member models to describe landscape evolution, we showed that if the rate of transport of sediment were assumed to be instantaneous (i.e. all sediment is transported out of the model domain) then the stream power model would be appropriate to describe catchment erosion. However, if it is instead assumed that the rate of sediment transport is not instantaneous, then we arrive at a model in which erosion scales with the rate of change of sediment flux, which itself is dependent on both linear and potentially non-linear slope-dependent terms. These two end-members can produce similar steady-state landscapes, as noted by a number of previous workers (e.g. Whipple and Tucker, 2002; Tucker and Whipple, 2002). However, as we demonstrate above, when perturbed by a change in conditions such as rainfall rate, they can produce very different landscape responses, which vary in terms of their style, magnitude and tempo. We explore the nature and implications of these responses below.

It is also important to stress that the catchment responses, and the predicted sediment fluxes out of these two model domains might be relevant variously to different erosional and depositional domains (c.f. Lague, 2014; Temme et al., 2017). A model of instantaneous sediment transport might be more relevant for suspended sedimentary loads, where transport times can be very small, while the transport model might be more appropriate for bedload dominated systems, even in cases where bedrock is clearly incised (e.g. Paola et al., 1992; Valla et al., 2010). Furthermore, given that these two models have different response times, it is possible that fine grained deposits might record signals of climate change differently from e.g. the gravel deposits within alluvial valleys. Below, we will therefore first discuss how the two model responses compare in terms of their response time, and place our results in the wider context of sediment routing system response to environmental change. Second, we will compare the model results with three records of change in sediment deposition during the Paleocene Eocene Thermal Maximum (PETM), a known and well-constrained period of rapid climate change. Finally, we will summarise the key implications from our results.

### 4.1 Response times as a function of model choice

Under certain parameter sets it is relatively straightforward to generate two landscapes, eroded by the transport or stream power model, that have similar elevation, slope, and area metrics (Figures 3 and 5). To find a path to break the apparent non-uniqueness of these solutions we have explored the transient sediment flux response out of the model domain for two end-member solutions to erosion. The first observation is that both models respond, at a first order, in a broadly similar way to a precipitation rate (climate) driver (Figures 9 and 10). Both models have a response that is an inverse function of the magnitude of precipitation rate change. Both models have a response that is related to uplift in an identical manner (Figure 13). However, the responses for catchments that are comparable in slope-area relationship and maximum elevation, but which are governed by different erosional dynamics defined by $c$, $k$, $m$ and $\delta$, actually display different response times by almost one order of magnitude (Figures 2 and 4).

We have demonstrated that models limited by their ability to transport sediment tend to have shorter response times to an increase in rainfall rate, and thus re-achieve pre-perturbation sediment flux values more rapidly compared to stream power

dominated systems, particularly when catchment length-scales are small (e.g. $< 100\,\text{km}$, Figure 10). These model observations suggest that the sediment fluxes from small alluvial catchments, even when captured in downstream depocentres, may be difficult to tie to changing climate parameters, unless depositional chronologies are exceptionally-well constrained (e.g. D'Arcy et al., 2017). Conversely catchments whose erosional dynamics lie close to the stream power end-member model, may

be well-placed to record longer-term climate shifts, but may be buffered to very high frequency variations in the climate driver (c.f. Simpson and Castelltort, 2012; Armitage et al., 2013). It is important to stress that the trend in response is asymmetric, by which we mean that both models show a faster response for a precipitation increase relative to a precipitation decrease (Figure 10). This is an important outcome, which has to-date not been widely recognised or investigate in field scenarios. In particular, it raises the prospect that for glacial-interglacial cycles, where characterised by wetter, cooler stadial periods and dryer, warmer

interstadials, the rapid climate recovery from peak glacial conditions typically seen in $\delta^{18}\text{O}$ records might be mediated by a longer landscape response time to this change. Conversely, physically slower boundary condition changes towards wetter conditions may give rise to faster landscape response times. We suggest that an exploration of these differences may be a promising avenue of future research.

Given that the response time is a function of the water flux exponent ($m$ or $\delta$), and that the water flux exponent for the

transport model is greater than that for the stream power model, there will be a cross over point where the stream power model responds faster than the transport model. This cross-over point is a function of the erodability coefficient $k$ and the transport coefficient $c$. In the scenario where we have tried to initiate the perturbation in precipitation rates from similar catchments, we find that this cross-over point is towards large reductions in precipitation rates (Figure 10). This implies that the transport model generally responds faster than the stream power model ($10^5$ to $10^6$ years), for examples in which the parameter combinations

used here produce similar steady-state landscapes.

For such conditions, the stream power model predicts a landscape response time to a change in precipitation of the order of $10^6$ yr, and this time is related to the precipitation rate to the inverse power of $m$ (Figure 10). The transport model predicts a wider range of response times of order $10^6$ to $10^5$ yr that is related to the precipitation rate to the inverse power of $\delta$, also in this case the response time is length dependent (Figure 10 and Table 2). It has been suggested that a transition from a

landscape controlled by detachment limited erosion (stream power model) to sediment transport at longer system lengths may explain the longevity of mountain ranges (Baldwin et al., 2003). This hypothesis is somewhat backed up by the analysis of response times for the transport model, as the response time increases with system length (Table 2) unlike the stream power model, which has a response that is only slightly modified by system length (Whipple, 2001; Baldwin et al., 2003). To-date physical constraints on landscape and sediment flux response times to climate changes in the geologic past are relatively

scarce (Ganti et al., 2014; Romans et al., 2016; Temme et al., 2017), because real systems are complex, they include internal dynamics such as vegetation and autogenic behaviour which are often omitted from model studies, and because of the need of stratigraphic archives to be complete with well-established chronologies (Allen et al., 2013; Forman and Straub, 2017). In principle, however, the dominant long-term incision process governing catchment behaviour fundamentally determines the sediment flux response and may itself help identify catchment erosional dynamics; we explore this question in section 4.2.

Finally, it is worth noting that the model response time has implications for the inverse modeling of river profiles. When river long profiles are inverted for uplift, erosion is typically assumed to be captured by the stream power model (e.g. Pritchard et al., 2009). Studies of continent scale inversion have found that the best fit value of $k$, for the stream power model, increases by two orders of magnitude to fit river profiles in Africa relative to Australia (Rudge et al., 2015). Such a large change in $k$ would result in a highly significant difference in response time from continent to continent, which in itself would imply that tectonic and climatic signals are preserved in landscapes and sediment archives over vastly different time periods (c.f. Demoulin et al., 2017). Such an outcome may reflect fundamentally differences in bedrock erodiblity (Roy et al., 2015), but alternatively could be satisfactorily explained by differing long-term erosional dynamics and sediment transport. These differences are enhanced in the case where $n > 1$ in the stream power erosion model.

## 4.2    Relevance of model responses to sediment records of climate change

To what extent do these model results, which start from similar steady-state topographies, help us to understand whether stratigraphic records of sediment accumulation through time do, or do not, reflect the effects of climatic change on sediment routing systems governed by differing long-term erosional dynamics? One motivation for this study has come from the increasing number of field and stratigraphic investigations of terrestrial sedimentary deposits, apparently contemporaneous with (and taken to record) known past climate perturbations, such as the Palaeocene Eocene thermal maximum (PETM), a hyperthermal event that occurred around 56 Ma. Stratigraphers often correlate changing stratigraphic characteristics to changing environmental boundary conditions in a qualitative way (c.f. Romans et al., 2016; Allen, 2017). However, to evaluate quantitatively how sediment routing systems respond to climate with reference to real examples, it is imperative to consider systems in which the timescales of erosion (or as a proxy, deposition) are known, stratigraphic sections are complete, and the driving mechanisms well-documented (c.f. Allen et al., 2013; D'Arcy et al., 2017).

To compare our model predictions with observations it is clear that we have to use the depositional record. Therefore, there is an implicit assumption that stratigraphy is a faithful recorder of erosion. It is however possible that climatic change will also alter processes that control sediment deposition, for example by altering how sediment partitions from transport into stratigraphy. By using estimates of the total volume of sediment deposited within the Escanillia Eocene sedimentary system in the Spanish Pyrenees, it has been demonstrated that climatic change can recreate observed change in grain size deposition (Armitage et al., 2015). This example of a close model-to-stratigraphic-observation prediction might be evidence that the stratigraphic record is a faithful record of change in sediment flux delivery to the depositional environment.

The PETM is arguably the most rapid global warming event of the Cenozoic, with a rise in global surface temperatures by 5 to $9\,^{\circ}$C (Dunkley Jones et al., 2010), forming a clear step-change in climate for which depositional records are well-constrained in a number of basins world-wide (Foreman et al., 2012). It is therefore a good example for high-level comparison with our model outputs. While the large magnitude glacial-interglacial cycles of the past 1 Myr are also plausible candidates to investigate these links in principle (c.f. Armitage et al., 2013), we note that many terrestrial records of sedimentation over ca. 100 kyrs, such as fluvial terraces and alluvial fans, have depositional chronologies that are often incomplete or reworked (D'Arcy et al., 2017; Demoulin et al., 2017).

The initial warming associated with the PETM event occurred at ca. 55.5 Ma, and may have been as abrupt as 20 kyr, with a duration of 100 to 200 kyr based on synthesis of $\delta$13C and $\delta^{18}$O records (e.g. Schmitz and Pujalte, 2007; Foreman et al., 2012). The event has been associated with clear changes to global weather patterns, for instance hydrogen isotope records suggest increased moisture delivery towards the poles at the onset of the PETM, consistent with predictions of storm track migrations during global warming (Sluijs et al., 411). This event has also been argued by an increasing number of authors to have produced a significant geomorphic and erosional impact based on sedimentary evidence, and its apparent effect on the global hydrological cycle and catchment run-off (e.g. Foreman et al., 2012; Foreman, 2014).

A clear response to the PETM is recorded within both the Spanish Pyrenees and the western U.S. in North America; however the responses are arguably not the same. At the onset of the PETM there is strong evidence for the contemporaneous increase in precipitation rates and the deposition of coarse gravels known as the Claret Conglomerate (Schmitz and Pujalte, 2007), in the Tremp Basin of the Spanish Pyrenees. In the western U.S. the PETM is marked by the deposition of well-documented channel sandstone bodies in the Piceance Creek and Bighorn Basins (Foreman et al., 2012; Foreman, 2014, e.g.). In the US cases, the deposits include coarse channelized sands, marked by upper flow regime bed forms, some of which are consistent with a synchronous increase in both water and sediment discharge. At Claret, where the style of sedimentation abruptly changes from a semi-arid alluvial plain to an extensive braid plain or megafan deposit, the conglomerate has a thickness of $\sim 10$ m while the total carbon isotope excursion (CIE) in the same section measures $\sim 35$ m (Manners et al., 2013).

### 4.2.1   Claret Conglomerate, Spanish Pyrenees

The Claret Conglomerate was likely deposited rapidly, representing a fast response to climate change. If we assume a constant rate of deposition, then the Claret Conglomerate accounts for roughly 30 % of the total duration of deposition for the CIE (170 kyr; Röhl et al., 2007), suggesting deposition occurred over a duration of up to 50 kyrs. Indeed, Schmitz and Pujalte (2007) argue that the deposition of this unit may have been markedly quicker than the conservative estimate above, perhaps taking less than 10 kyrs, based on their detailed comparison of $\delta^{13}$C and $\delta^{18}$O records at the field site, compared to marine records of the excursion. Therefore unless there is a major unconformity within the CIE, the implication is that the erosional system responded rapidly at this particular field site, in 10 to 50 kyr to a significant shift in climatic conditions. These values suggest sedimentation rates of up to $1 \, \mathrm{mm \, yr^{-1}}$. If such a sedimentation rate had been sustained for the duration of the deposition of the Tremp Group (Maastrichtian – end Palaeocene), the sediment thickness would be $> 15$ km. This is an order of magnitude more than actually observed (Cuevas, 1992), and therefore would suggest that sediment fluxes increased dramatically at the PETM.

Erosional source catchment areas were likely $< 100$ km in length, given the palaeo-geography of the Pyrenees at the time (Manners et al., 2013). The very short duration of the erosional response, which is required for the sediments to be transported and deposited in a timescale of ca. $10^4$ years is therefore difficult to model within a stream power (advective) end-member model for catchments of this scale (Table 2), although a version of such a model has been recently used to explore the controls on the evolution of later Miocene megafans in the northern Pyrenees (e.g. Mouchené et al., 2017). To use the stream power model would require a significant increase the bedrock erodibility parameter, $k$, within the model (by greater than one order

of magnitude), implying slopes and topography in the palaeo-Pyrenees that were highly subdued. In contrast, the sediment transport model more easily reproduces the documented response timescales given an increase in precipitation, is consistent with the volumetrically significant export of bedload transported gravel clasts, and therefore honors the independent field data more effectively. We also note that the transport model displays a response time that has a stronger dependence on precipitation rate change, and has a greater amplitude of perturbation (e.g. Figure 9). We therefore suggest the erosional pulse that led to the deposition of the Claret conglomerate is most appropriately modelled as a diffusive system response to a sharp increase in precipitation over the source catchments of the developing Pyrenean mountain chain at that time.

### 4.2.2  Sandstone Bodies in the Piceance Creek and Bighorn Basins, Western U.S.

The time-equivalent sections in the Bighorn and Piceance Creek basins of the western U.S. also provide clear evidence of anomalous sedimentation at the PETM, however in this case the duration of deposition is somewhat longer, $> 100\,\mathrm{kyrs}$ (Foreman et al., 2012; Foreman, 2014). Here the deposits are of smaller grain sizes, with the Boundary Sandstone sequence in the Bighorn basin being made up of fine to coarse sand with little gravel (Foreman, 2014). In the Piceance Creek basin, the PETM section documents the rapid progradation of coarse-grained sands, consistent with greater discharges, over silty underlying strata and in that sense these observations also match sediment transport model predictions for rapid increases in sediment flux, driven by enhanced precipitation (Foreman et al., 2012). However, it is notable the documented changes in fluvial style persisted beyond the PETM and we therefore suggest that fast response of system to the increase in precipitation, but the persistence of coarser grain sedimentation as the climate presumably dried and cooled may indeed reflect the marked asymmetry in sediment flux responses to wetting and drying noted in Figure 9.

In contrast, the Bighorn basin boundary sandstone sediments are contained within the PETM time period, and indicate uniform flow depths and widths during this time, while also being coarser than the underlying horizons. Moreover, proxy data suggest a net decrease rather than increase in precipitation (Foreman et al., 2012; Foreman, 2014). As the authors note, while progradation of such coarse-grained facies could be represented as a diffusive process driven by increasing rainfall-driven discharge (Paola et al., 1992; Armitage et al., 2011), this is apparently inconsistent with the sedimentological characteristics of the deposit. Although sediment fluxes are not explicitly reconstructed in this work, this response apparently requires greater volumes and grain sizes of sediment delivered despite lowered rainfall conditions, and is thus difficult to capture in either of the end-member models used here. The authors argue for the prefential removal of fine-grained flood plain deposits, speculatively linked to changing vegetation and the reduced cohesion of overbank sediments. Consequently, while two of the PETM sections considered here are broadly consistent with landscape responses governed by a sediment transport model, some depositional stratigraphies are not immediately consistent with either end-member model, and may reflect important complexity such as the effects of vegetation lacking from simple model solutions.

### 4.3  Summary and model limitations

In this discussion section we have considered the implications of our model outputs, both generally for interpreting sediment routing system response to boundary condition change and specifically in the context of the well-studied PETM event.

While the sediment flux response of the models to a change in precipitation are at a first order level broadly similar, there are four key differences to highlight. First, starting from the same initial conditions, the sediment transport model appears to be more sensitive to precipitation change than the equivalent stream power model. It therefore is a good candidate where rapid catchment-wide responses are recorded to e.g. a climate change event, as we argued for the PETM Claret conglomerate in the Spanish Pyrenees. Second we note that in both model cases there is a quicker response to wetting than drying event, something which has not been well-established or demonstrated from field observations. Nevertheless we argue that field data sets, including PETM studies may already have recorded this asymmetry, although it may not have been recognised as such. Third, the sediment transport model has greater magnitude of peak sediment flux and is particularly sensitive to catchment size. Finally we note that response time in both models is a function of uplift rate for $n > 1$, which means in such cases, perhaps counter-intuitively, that the more perturbed the system the faster it responds (c.f. Whittaker and Boulton, 2012).

However, it is important to recognise that in deriving these two classic end-member models we have simplified landscape evolution considerably. We acknowledge that change in the model parameters, $c$, $k$, $m$ and $\delta$, will alter the response times depicted here (c.f. Armitage et al., 2013). However, in order to compare the two models we have specifically used values of $c$, $k$, $m$ and $\delta$ that generate comparable model landscapes, and we then changed the precipitation rate to understand the form of the model response. No model incorporates all the complexities that characterize sediment routing systems from source to sink (c.f. Allen, 2017) and the act of simplification inherent in considering erosional end-member models necessitates that in arguing for the applicability of one over the other, we simply consider the broad styles of behavior suggested by model outputs. We do not, for example, consider autogenic behaviours (e.g. Forman and Straub, 2017), nor do we consider coupled issues of vegetation turn-over in response to climate change, which may a play an important role in examples such as the Bighorn basin considered here (c.f. Foreman, 2014). Nonetheless, a significant finding of this work has been the clear asymmetry in response time of these end-member models in terms of a wetting event (faster) compared to a drying event (slower). This implies that aridification events are harder to preserve in the sedimentary record, not only because they are typically associated with reduced sediment fluxes, but also because the timescale of landscape response may be $> 10^6$ years.

## 5   Conclusions

Deterministic numerical models of landscape evolution rest on fundamental assumptions on how sediment is transported down system. The stream power law is based on the assumption that all sediment generated is transported instantaneously out of the landscape. transport models assume that there is an endless supply of sediment to be transported. The existence of knickpoints within river long profiles, assumed to be produced by a system pertubation such as a base level, has been used to provide evidence in support of the stream power law in upland areas (e.g. Whipple and Tucker, 1999; Snyder et al., 2000; Whittaker et al., 2008). Knickpoints however can likewise be a result of changes in lithology (Grimaud et al., 2014; Roy et al., 2015) and are certainly not a unique indicator of erosion dynamics (e.g. Tucker and Whipple, 2002; Valla et al., 2010; Grimaud et al., 2016). In this contribution we therefore attempted to understand of the sediment flux signal out of the eroding catchment may generate a distinguishable difference between the end-member models in term of a response to a change in run-off. This idea is

motivated from field observations of past landscape responses to climate excursions, such as the PETM, which are manifested in the rapid deposition of coarse sedimentary packages in terrestrial depocentres (Armitage et al., 2011; Foreman et al., 2012).

Both models suggest that the response time of landscape to change in precipitation rate has a proportionality of the form of a negative power law (equations 15 and 16). The key difference is in the value of the exponent. For the stream power model, the exponent must be less than one in order to match the observed concavity of river profiles. In contrast, for the transport model the exponent on the precipitation rate must be greater than one in order to generate a river network (Smith and Bretherton, 1972), and to generate the observed concavity of river profiles. This results in the transport model responding more rapidly to an increase in precipitation rate in comparison to the stream power law model (Figure 10). In contrast, the stream power model is faster to respond to a reduction in rainfall rate. This is fundamentally because the response time of this model is more weakly a function of precipitation than the sediment transport model. Significantly, therefore, our results show that there is a fundamental asymmetry in the response of both models to a climatic perturbation, with the response time to a drying event longer than that to an increase in rainfall. In general terms, the magnitude of the response to a change in precipitation rate appears greater across the range of model space investigated here for the sediment transport (diffusive) model solutions, while for the stream power (advective) model, the magnitude of the sediment flux perturbation is smaller, but is more localised within the catchment with respect to knickpoint retreat.

While this study does not address whether or not these sediment flux signals will be preserved in the stratigraphic record, a problem that fundamentally rests on the availability of accommodation to capture the eroded sediment (c.f. Allen et al., 2013; Whittaker et al., 2011), it does suggest that landscapes governed by these simple erosional end-members should be sensitive to climate change; and moreover that there are some important diagnostic differences between their sediment flux responses to an identical perturbation, including the amplitude, timescale and locus of the erosional response. Using published stratigraphic examples, we suggest that the timescales and magnitude of coarse sediment deposition in the Spanish Pyrenees at the time of the PETM are best described using the diffusive transport model end-member. Moreover, we argue that these model end-members allow us to constrain the range of likely sediment flux scenarios that precipitation changes may generate, and that numerical models, in conjunction with a range of field and independently-constrained proxy data sets are best placed to tease apart when and in what circumstances climate signals are likely to have been generated in erosional catchment systems, which fundamentally determines whether they can be subsequently *captured* in sedimentary depocentres downstream.

## 6  Code availability

The 1-D solution to the transport model is available from John Armitage (armitage@ipgp.fr). The 1-D solution to the stream power model is available from Benjamin Campforts (benjamin.compforts@kuleuven.be). Fastscape is available from Jean Braun (GFZ Potsdam) by request. The 2-D solution to the transport model was developed by Guy Simpson (University of Geneva) and is available as part of Simpson (2017).

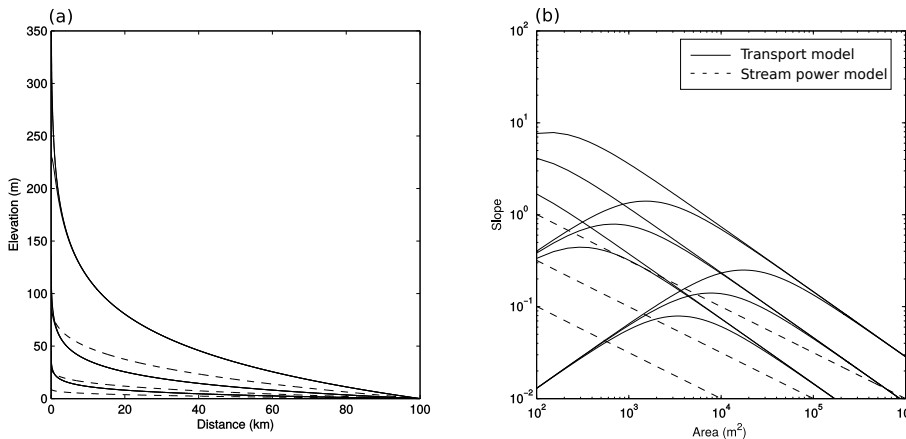

**Figure 15.** (a) Steady state profiles of elevation against down-system length, and (b) the slope of the profile plotted against the drainage area assuming that area $a = x^p$ where $p = 1/h = \sqrt{2}$ and $h$ is the Hack exponent. Dashed lines are for the stream power law (Equation 12) with $m = 0.5$, and $k = 10^{-4}$, $10^{-3.5}$, and $10^{-3}$. The solid lines are for the transport model (Equation 8 with $n = \sqrt{2}$, $\kappa = 10^{-3}$, 1 and $10^3 \, \mathrm{m^2 \, yr^{-1}}$, and $c = 10^{-6}$, $10^{-5.5}$, and $10^{-5}$.

## Appendix A: Steady state 1-D profiles

The solution to the one dimensional stream power law (Equation 12) assuming that at the end of the catchment at $x = L$ elevation $z = 0$ and $mp \neq 1$ is,

$$z_{sss} = \frac{U}{mk\alpha^m (mp - 1)} \left( x^{(1-mp)} - Lx^{(1-mp)} \right) \tag{A1}$$

5   and for the case where $mp = 1$ this simplifies to,

$$z_{sss} = \frac{U}{k\alpha^m} log_e(L/x) \tag{A2}$$

For the transport model (Equation 8) there is an exact solution for the case that $\delta p = 2$, which assuming at $x = 0$, $\partial_x z = 0$ and at $x = L$, $z = 0$ is,

$$z_{sst} = -\frac{UL}{2\kappa D_e} \left( log(D_e x^2 + 1) + log(D_e + 1) \right) \tag{A3}$$

10   where,

$$D_e = \frac{ck_w \alpha^{2/p} L^2}{\kappa}. \tag{A4}$$

For other values of $\delta p$ the steady state solution is solved for numerically, where Equation 8 is solved using the finite element method with linear weight functions. We use a non-uniform 1-D nodal spacing, where the spatial resolution is increased with increasing gradient. The numerical model is bench-marked against the analytically solution for the case where $np = 2$.

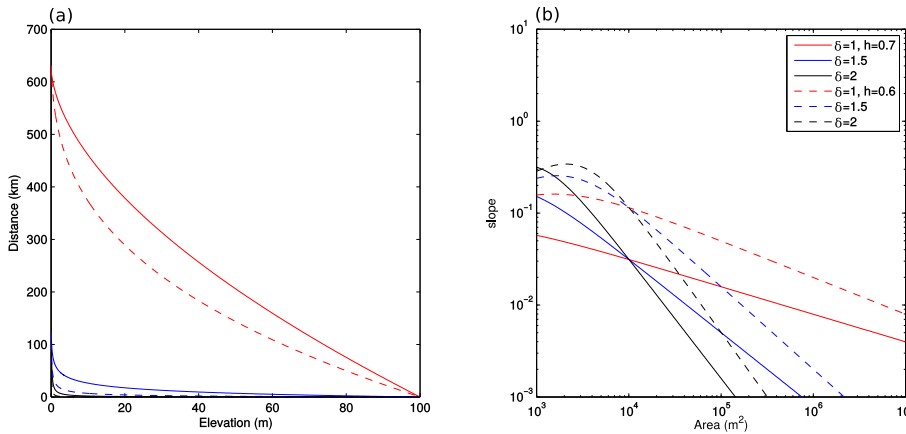

**Figure 16.** (a) Steady state elevation and (b) slope-area relationship for the numerical solution to 1-D sediment transport (Equation 8), where the area, $a$, is taken to be related to distance $x$ by $a = x^p$ where $p = 1/h$ and $h$ is the Hack exponent. Red lines are for the case where $n = 1$, blue lines for $n = 1.5$ and black lines for $n = 2$. Solid lines are for $h = 0.7$. Dashed lines are for $h = 0.6$.

The steady state solutions are plotted in the case that $\delta = p = \sqrt{2}$ and for reference the stream power model solution with $m = 0.5$ and $p = \sqrt{2}$ (Figure 15). Such a value of $p$ assumes that $h \sim 0.7$, which is towards the higher end for observed Hack exponents and that the river catchment is very elongate. When plotting the logarithm of the model slope against drainage area (Figure 15b), where area is given by $a = x^p/k_w$ and assuming $k_w = 1$, for the simple stream power law derived here the slope

area exponent $\theta = -m$. The value of the dimensional constant $k$ has no impact on the slope area exponent as expected. The transport model likewise creates river long profiles that have on average a negative curvature. For small values of $x$ there is however a region of positive curvature where $\kappa > ck_w\alpha^\delta L^{\delta p}$. For the slope area analysis this leads in there being a positive gradient in the trend for small catchment areas. This relationship subsequently has a negative slope for larger catchments. The point of inflection is dependent on the value of $D_e$, where for smaller values of $\kappa$ the region of positive gradient is reduced.

There is therefore a critical catchment area that is dependent on the diffusive term $\kappa$. After this critical point the slope area relationship becomes negative. At distances down-system, where the upstream area is greater than this critical area, the gradient $\theta = -0.88$. $\theta$ is insensitive to the coefficient $c$ as would be expected.

The range of gradients found for river catchments for this type of slope area analysis, usually referred to as concavity, generally lies within the range $\theta = -0.35$ to -0.70 (Snyder et al., 2000; Wobus et al., 2006). It is trivial to find the values of

$m$ for the steady state solution to the stream power law such that fits such values of $\theta$. To further explore how $\theta$ depends on $\delta$ and $p$ within the transport model we solve Equation 8 numerically for $\delta = 1$, 1.5 and 2 while keeping $h = 0.7$ or 0.6 (Figure 16). The result is that $\theta$ varies from -0.3 for the case of $\delta = 1$ to -1.31 for $\delta = 2$. The values of the gradient for the slope area analysis for $1.4 < p < 2$, where we are assuming $d = 1$ and hence $p = 1/h$, are displayed in Table 3. For the transport model the slope is dependent on both $\delta$ and $p$.

Clearly there exists a combination of $\delta p$ that is equally capable of fitting the observed river long profile. Furthermore, for the transport model the slope is a function of the Hack exponent $h$ (and therefore $p$) and the choice of $\delta$. This because of the

**Table 3.** Gradient, $\theta$, of the slope vs. area trend at steady state for 1-D sediment transport (Equation 8, Figure 16).

| $\delta$ | $p = 1.40$ | | $p = 1.67$ | | $p = 2.00$ | |
|---|---|---|---|---|---|---|
| | $c$ | $\theta$ | $c$ | $\theta$ | $c$ | $\theta$ |
| 1 | $10^{-6}$ | -0.50 | $10^{-5}$ | -0.40 | $10^{-4}$ | -0.30 |
| 1.5 | $10^{-8}$ | -1.01 | $10^{-7}$ | -0.91 | $10^{-6}$ | -0.81 |
| 2 | $10^{-10}$ | -1.51 | $10^{-9}$ | -1.41 | $10^{-8}$ | -1.31 |

diffusivity term that leads to positive curvature and rounded 1-D profiles (Figure 15b and 16). The magnitude of the water flux term within the transport equation (Equation 8) is dependent on how much water the river network captures, which is in turn a function of how elongated the catchment is.

The positive slope area relationship for the transport model for small catchment areas, see Figure 15b and 16b, has been previously explored in Willgoose et al. (1991). The gradient of the relationship between slope and catchment area is dominantly a function of the exponent $\delta$ within Equation 6. The value of this exponent is likely within the range of $1 < \delta < 2$ depending on the bed-load transport law assumed (Armitage et al., 2013). If the observations of trunk river slope against catchment area are representative of a landscape at steady state, then for the smaller range of $1 < \delta \leq 1.5$, a realistic catchment topography can be generated.

*Acknowledgements.* We would like to thank Guy Simpson (University of Geneva) for sharing his numerical model that solves the Smith and Bretherton (1972) equations in two dimensions. We thank Tom Dunkley Jones (University of Birmingham) for discussions on the duration of conglomeratic deposition duing the PETM, Gareth Roberts (Imperial College London) for discussions on slope exponents and Jean Braun (GFZ Potsdam) for sharing his numerical model Fastscape. This work initiated under funding by the Royal Astronomical Society through a research fellowship awarded to John Armitage while he was at Royal Holloway University of London

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
