# Peer review of "NUMERICAL MODELLING OF LANDSCAPE AND SEDIMENT FLUX RESPONSE TO PRECIPITATION RATE CHANGE"

_Earth Surface Dynamics, 2017_

## Referee Comment (RC2) · Anonymous Referee #2 · 28 Aug 2017

General Comments: The objective of this paper was explore end-member models of landscape evolution with the goals of finding how changes in model forcing and model assumptions control model response time and whether changes in sediment flux during a perturbation is diagnostic of the end member models. The motivation for the study was whether the end member models can constrain sediment fluxes to depositional basins and allow the interpretation of past climate signals from the sedimentary record. Their main findings were that transport-limited models have a faster response time to a change in precipitation rate and sediment flux is higher in transport limited models. In my opinion, one of the most interesting parts is the asymmetry in the response times of transport limited models, with a longer response time to a drying event than to a wetting event, but this result was not discussed in detail. Overall, the qual-

ity of the paper including the motivating questions, the derivation of model equations, and the explanation of model set up (in appendix B) was good. I strongly suggest that the material in appendix B be moved to the Methods section. This will really improve the reading experience. The results were adequately explained, but at times the text was dense with much discussion about the results without references to appropriate figures. The weakest section of the paper was the discussion and conclusions. The authors conclude that a sedimentary record of the PETM is best explained by using a transport-limited model because this model has a faster response time. But a faster response time for one model doesn't rule out another model. I would like to see a more robust support of the sediment transport model in making this conclusion. Second, the authors favor the transport-limited model to explain a particular sedimentary record of the PETM because the instantaneous transport of bedload (assumed by stream power model) is not justified for specific case cited in the Pyrenees. But I don't think anyone would argue that stream power model is justified, so this weakens potential impact of paper. In the discussion, I think the paper could benefit from a more generalized framework of the circumstances under which the findings of this study are relevant. For example, what is needed in the field/sedimentary stratigraphy to distinguish between a landscape that was depositing under a detachment-limited vs. a transport-limited system? I think that exploring end member models to identify diagnostic characteristics of each in transient state is a worthy goal, but the authors could do a better job articulating how the diagnostic characteristics they have identified are useful in a larger context outside of interpretation of sedimentary records.

Specific Comments:

P1L19-21: Conclusion that rapid response in sedimentary basins more easily explained by using transport model for two reasons: 1) this model has a faster response time (Is this a new finding?) and 2) instantaneous transport of bedload (assumed by stream power model) is not justified for specific case cited in the Pyrenees. But I don't think anyone would argue that stream power model is justified, so this weakens potential impact of paper.

P2L6: "a series of experiments": give very brief description of experiments. Also examples of real catchments responding to changes in precipitation would be useful to give readers an broader framing for your work.

P2L15: Can you also point readers to your own modeled examples of landscapes that were created by transport model and stream power model, but are indistinguishable from each other at steady state.

P2L23: Mentioning that river profiles can be inverted to extract climate history at this point seems to bog down the explanation of differences between advective and diffusive dominated systems in transient state.

P2L27-31: Transport/diffusive models do not produce knickpoints in transient state. Pointing out that knickpoints occur for reasons other than a transient state in advective-dominated systems doesn't change that, nor does it support the motivation for the study in the following lines (L31-33). That said, I think exploring model end member behavior is a worthy goal.

P3L7: typo "dirven"

P3L7-8: Be more specific here about what experiments show about response time to a perturbation. It's self-evident that there will be a response time for systems to return to steady state.

P3L10: Make it clear that this is the new piece this study adds to the existing body of knowledge.

P3L20-21: Be clear and consistent throughout the paper with terminology when referring to advective/stream power law/detachment limited and diffusive/sediment transport model/transport limited. These are used interchangeably throughout the manuscript. I recommend explaining the meaning of all three descriptions for each endmember model early in the paper, then using one of the terms for the rest of the paper.

Figure 1: include erosion, E, in the figure.

P4L10: Why ask the question of if mass transport is appropriate at continent scale when this paper doesn't answer that question. It seems to me the paper addresses the question of if advective transport is appropriate for all models with changing boundary conditions.

P4L13-14: reference here, e.g. Davy and Lague 2009.

P5L1-8: This discussion of mass transport in suspension seems unnecessary here and unrelated to the point of the paper.

P5L12: define physical meaning of coefficient kp, bedrock erodibility, for readers who are not familiar with erosion models.

P5L20: To this point, the derivation is good, easy to follow for the most part, except authors need to define kp, as noted above.

P5L24-26: alpha (precipitation rate, I found later) is not defined here and makes this section very confusing.

P5L27: more specific definition of kw, width coefficient?

P6L1-5: Before you launch into derivation of transport-limited model, give a few more lines to discussing what this means physically, in a natural system. Also here explicitly say this is the transport-limited erosion equation we use.

P6L23: a fourth name/way of describing stream power as a kinematic wave equation. This is obvious to people who are immersed in the world of erosion models, but many readers are not. So again, be careful and consistent in the terminology used to describe the two end member models.

P6L25: can you reference a figure that shows a migrating knickpoint?

P6L25-27: mentioning shock wave seems unnecessary and distracting to me. General comment about derivations: there is lots of discussion of various exponents, but I would like to see explanations of the link exponent values with natural systems where possible.

P7L16-19: why do you use different grid sizes for the two models? Does this affect the outcome of the models?

P7L22-24: This is confusing on the first read through. Explain more or reference figure that shows this relationship (maybe Figures 12 or 14)

P7L25: you've switched from deriving stream power model first, then transport model to discussing transport model first, then stream power. It would help readers follow if you kept the same order throughout the paper, but I recognize that's difficult and perhaps not always possible. Methods: I strongly recommend moving Appendix B to the methods section so readers have a better idea what you're doing and what these models look like before discussing results. This is very important and will help the readability of the paper.

Figures 2 and 3: These figures are not high impact by themselves. Suggest combining these figures and then give them clear titles indicating which end member model results are shown.

P8L16-17: this was the first time it was clear to me that the heart of this paper is model response to changing precipitation rates. Emphasize this more clearly in the methods section.

P9L2: Sentence that begins, "The response to a reduction in precipitation...." is confusing because it's not specific and is a bit of a run on. Needs punctuation or splitting into two sentences to make it clearer.

P9L3-6: Very long run on sentence. This sentence says "importantly, however, it changes the model elevation...". Reference a figure where readers can see this change in elevation with changing c.

P9L4: Be specific about where in appendix A readers should look

P9L10-11: Reference one of your figures that shows where these three different sets of parameters result in similar model topography, e.g. figure 14.

Figure 4 and 5: These need titles to make it clear that one is the transport model and one is the stream power model.

Figure 5: indicate knickpoint on figure 5a

Figure 6: this figure must have titles indicating that one is the transport model and one is the stream power model.

P12L3-5: It seems to me that it's at least intuitively known that aggradational (drying) events happen more slowly than an erosional (wetting) event, but I couldn't point to a reference for this. If there aren't many studies that show this, I think this makes a very interesting point. If there are studies, they should be cited.

Figure 7: include in caption these results apply to catchments where L=100km only. Should also make clear where the data to make this figure comes from. Is it just a line through your model data? If made from model data, these should be indicated with points and justification given for how the line was drawn through the model data.

P13L1-2: include reference for knickpoint celerity models, e.g. Berlin and Anderson 2009, JGR

P14L2-3: This seems important that response time takes twice as long when L=500 km compared to L=100km. I suggest a figure showing this difference in response time.

P14L12-13: explain what this empirical evidence of 0.5<h<0.7 means in a physical system.

Figure 9: show points where you have data from model runs.

P16L2-3: reference figure 7 that shows how models respond to precipitation rate.

P16L6: reference figure that shows similarities between landscapes created by transport models and advective models.

P16L7-9: I appreciate the goal, but what constraints are needed to make this useful? See comment below on conclusions.

P16L13: reference figure that shows difference in response times.

P17L1-2: this asymmetry is interesting. Reference figure that shows it.

P17L7-8: Can you say more about catchment response time for the sediment transport model? For example, when is this information useful for evaluating sediment records? I think this point needs to be discussed more thoroughly.

P18L2: At Claret: mention this is the site in the Pyrenees.

P18L6-10: confusing. What is the justification for comparing paleosols in the Bighorn Basin with paleosols in Claret? Perhaps too much detail in this section to get to the point of rapid deposition.

P19L12-14: I don't see why one would attempt explain evolution of a megafan with a stream power model in the first place. This is a bit of a strawman argument.

P18L20-24: Run on sentence.

P19L1-3: Again, it seems obvious that bed load transport is more easily described by a diffusive/transport limited model rather than the stream power model. It would be useful to reference studies where authors have used stream power to model bedload dominated systems. Relevance to sediment record and climate change: This section should include a more generalized framework of the circumstances under which the findings of this study are relevant. What is needed in the field/sedimentary stratigraphy to distinguish between a landscape that was depositing under a detachment-limited vs. a transport-limited system? For example, magnitude and duration of precipitation change, magnitude of sediment flux. It's also important to include an expanded discussion of why the sedimentary record at Claret is difficult to explain with an advective model. How much does precipitation have to change in the time period of deposition for an advective model to work? Is this anywhere close to reality? Generally, I want more to back up the conclusion that the transport model explains deposition in the sedimentary basins simply because it has a faster response time.

P19L10-12: I don't think that noting that knickpoints are not a unique indicator of erosional dynamics is helpful for understanding the motivation of this study.

P19L23-24: This is one of the most interesting outcomes of the paper. I would suggest discussing this and the implications of this effect on interpretation of the sedimentary record.

P20L1-4: nicely summarized objective of the paper. I would like to see this in the abstract also Figure 10: Figure needs a legend for the lines Figure 11: It's not immediately clear why both Figure 10 and Figure 11 are necessary. It's explained further down, P22L3-7, but figure 11 is initially confusing.

P22L12-13: show example of how slope-area relationship is sensitive to river network in T-Lim case. Or remove this line as it's not relevant.

P22L14-20: The point of this paragraph is unclear even after reading several times. Rewrite and reword.

Appendix B: As noted above, I think appendix B should be included before results are discussed. It would make the paper flow much more smoothly.

Figure 12: Labels on the figures: transport limited models.

P24L5: typo, should read steady state.

Figure 14: Figure needs label: stream power models. Also, there is an error in the caption. The caption currently says sediment transport model, should read stream power model.

**ESurfD**

---

## Author Comment (AC1) · 2 Oct 2017

We would like to thank Laure Guerit for taking the time to review our manuscript. Below we respond to the major comments raised:

(1) This work would benefit from some rewriting and reorganization to make the purpose of the authors more clear and easier to follow: some paragraphs could be reorganized and/or developed in particular to better highlight the state of the art in the domain (Introduction) or to develop some important aspects of the work (Method).

We have rewritten the introduction and methods section. In the introduction we have now tried to better place the numerical models within our current understanding of system response to precipitation rate change. However, rather than move the main

equations for the stream power model and the transport model, we would prefer to keep them within the methods section. This way they follow logically from the derivation and basic assumptions. To a non-specialist it can be a bit distracting to launch straight into the stream power law equation, for example.

The methods section has been reorganized. We now include Appendix B within the methods, which allows us to demonstrate how the transport model and stream power model evolve with out a perturbation, and where their predicted slope-area metrics are similar.

A particular point was raised in the review relating to the different grids used for the transport and stream power models. In the transport model, we us a triangular mesh and for the stream power model we use a rectangular mesh. The model resolutions are therefore not exactly the same.

The transport model solves the equations using a finite element approach, and a triangular grid is 2nd order accurate and an appropriate. The stream power model uses a finite difference approach with a rectangular grid, which is also 2nd order accurate. It is known that resolution can effect the drainage patters predicted in these sorts of models (see Schoorl et al., ESPL, 2000). For this reason we attempted to get the model resolutions to be similar. We also checked that the sediment flux was not influenced by grid resolution for both models. We are not convinced however, that this detail requires to be discussed within the manuscript.

The second point relating to the methods was the choice of perturbing the model at either 10 or 5 Myr. The non-linear models were perturbed sooner, at 5 Myr, because the 1D models were at steady-state by 5 Myr, and we did not see reason to run them for longer before the precipitation rate was changed. This point has been clarified within the methods section.

(2) The second major comment related to how we only changed the final precipitation rate, and held the initial precipitation rate fixed at 1 m/yr for all models. It was suggested

that we explore how the model response is a function of the initial precipitation rate.

This was a good suggestion. We have added a new section exploring how both models, in their linear and non-linear forms respond to a change in initial precipitation rate where the final precipitation rate is held fixed at 1 m/yr for all scenarios (Figure 1). We have found that for the transport model, the response time is sensitive to the initial precipitation rate (Figure 1a). However the proportionality does not fit a power law as was found for the relationship between response time and the final precipitation rate. Furthermore, the change in response time as the initial precipitation rate changes is of an order of magnitude, suggesting that response times are predominately a function of the final precipitation rate (Figure 1a).

For the stream power model we find that the response time is not a function of the initial precipitation rate (Figure 1b). We believe this difference is in how the transport model responds directly across the whole catchment, so that slopes at the uppermost reaches of the catchment still have a memory of the previous precipitation rate. For the stream power model, the erosion increases bottom-up and so there is no memory of the previous slope and topography.

(3) The third major comment relates to the discussion, that we are overestimating the duration of the Claret Conglomerate deposition. Furthermore, Laure Guerrit asks what evidence there is that sediment fluxes changed.

We have incorporated the Schmitz and Pujalte 2007 reference which Laure suggests; this paper suggests the timescale of deposition of the Claret conglomerate could be less than 10 kyrs. We have deleted any reference to palaeosols in the Bighorn basin and have simplified the calculation of the duration of the PETM conglomeratic deposition. We know that the sedimentation rates and sediment fluxes must have increased – Taking the Schmitz and Pujalte rates which Laure advocates suggests a sedimentation rate of 1 mm/yr – this is an order of magnitude bigger than for the rest Tremp Group and we now mention this explicitly.

(4) Laure Guerit asks if we could plot the stream power model response as a function of precipitation rate for other values of m.

The relationship between response time and m has been published elsewhere (e.g. Wipple, 2000). We are not convinced it is necessary to add further plots of response time for the stream power model in this manuscript, particularly when we wish to focus on model cases where the landscapes are similar.

Finally, we have addressed all the minor comments in the revised manuscript.
* * *
[Figure]

**Fig. 1.** Log-log plots for the transport model and the stream power model in 1-D for a step change in precipitation rate, where the initial precipitation rate is changed.

---

## Author Comment (AC2) · 2 Oct 2017

John J. Armitage

armitage@ipgp.fr

We would like to thank the reviewer for taking the time to review our manuscript. Below we respond to the major comments raised:

(1) Conclusion that rapid response in sedimentary basins more easily explained by using transport model for two reasons: 1) this model has a faster response time (Is this a new finding?) and 2) instantaneous transport of bedload (assumed by stream power model) is not justified for specific case cited in the Pyrenees. But I don't think anyone would argue that stream power model is justified, so this weakens potential impact of paper.

First, we believe that the faster response time for the transport model is a new find-

ing. Second, the stream power model is used to invert river profiles across whole continents (e.g. Rudge et al., G-cubed, 2015), and has been used, for example, in explaining drainage reorganization in catchments draining off the Appalachians into the Eastern North American Margin (Willett et al., Science, 2014). Moreover, it has been recently used to model erosion and the evolution of Miocene Pyreneean megafans (e.g. Mouchene et al, 2017, published in Earth Surface Dynamics this year). We contend that the stream power model is used to model erosion across continents with out much thought for the processes it represents. We also suggest that the treatment of erosional end of the catchment – e.g. do we use a stream power model to generate sediment fluxes is not the same as treating the depositional end – Qs from a stream power model could easily be fed into any type of depositional or stratigraphic model or reconstruction.

To address this point we have substantially rewritten the discussion section about comparison to field sites to make this point as clear as possible.

(2) "a series of experiments": give very brief description of experiments. Also examples of real catchments responding to changes in precipitation would be useful to give readers an broader framing for your work.

We have reworded the paragraph to state:

"In laboratory studies, a series of experiments where granular piles of a length scale of order of centimetres are eroded due to surface water, have demonstrated that a change in precipitation rate leads a period of adjustment of the landscape topography until a new steady-state is achieved (e.g. Bonnet & Crave, 2003; Rohais et al, 2011). These experiments use a mixture of granular silica of a mean diameter in between 10 and 20 $\mu$m, that is eroded by water released from a fine sprinkler system above. Given the complexity of these experiments, unfortunately there have been insufficient different precipitation rates studied to fully understand how the recovery time-scale varies as a function of precipitation or other parameters. In this contribution we will focus on

this transient period of adjustment to a perturbation in precipitation rates, and using numerical models will attempt to evaluate how the response time varies as a function of the model forcing.

(3) Transport/diffusive models do not produce knickpoints in transient state. Pointing out that knickpoints occur for reasons other than a transient state in advective-dominated systems doesn't change that, nor does it support the motivation for the study in the following lines (L31-33). That said, I think exploring model end member behavior is a worthy goal.

We take the reviewers point and have therefore deleted the text that discusses how knickpoints may form in the absence of a stream power model.

(4) Be more specific here about what experiments show about response time to a perturbation. It's self-evident that there will be a response time for systems to return to steady state.

As mentioned in the response to point 2, we now include a more specific description of the experiments.

(5) Make it clear that this is the new piece this study adds to the existing body of knowledge.

We have added "Sediment flux response times for the advective stream power law have been previously characterised by Whipple (2001) and Baldwin et al. (2003), and for the transport model they have been studied by Armitage et al. (2011) and Armitage et al. (2013), but not systematically or using 2-D models. Furthermore, to our knowledge no comparison between the transport model has been previously made."

(6) Be clear and consistent throughout the paper with terminology when referring to advective/stream power law/detachment limited and diffusive/sediment transport model/transport limited. These are used interchangeably throughout the manuscript. I recommend explaining the meaning of all three descriptions for each endmember

model early in the paper, then using one of the terms for the rest of the paper

We have made sure the terminology is consistent, where we refer to either the transport model or the stream power model.

(7) Why ask the question of if mass transport is appropriate at continent scale when this paper doesn't answer that question. It seems to me the paper addresses the question of if advective transport is appropriate for all models with changing boundary conditions.

Fair point. We have deleted the question. However in this we are not changing boundary conditions, rather the precipitation rate which impacts the transport coefficients.

(8) This discussion of mass transport in suspension seems unnecessary here and unrelated to the point of the paper.

We disagree. If mass transport is not in suspension then it is along the bed, and not rapid. Therefore, if mass is not transported as a suspended load the idea of instantaneous mass transport, which is implicit in the derivation of the stream power law, is wrong and hence the model is nonsense.

(9) General comment about derivations: there is lots of discussion of various exponents, but I would like to see explanations of the link exponent values with natural systems where possible.

We could relate the stream power parameters to nature, but not as a unique parameter value set, rather as a broad parameter value space (see eg. Croissant and Braun, 2014 ). However, there is an inherent trade off between the constants and exponents in both models that makes relating the m to natural landscapes challenging. Unfortunately we are of the opinion that desire to see a link between the exponents and natural systems is beyond us.

(10) Why do you use different grid sizes for the two models? Does this affect the outcome of the models?

**ESurfD**
We use a traingluar grid to solve the transport model as this is appropriate for the finite element implementation of the numerical solution. This makes having the same grid size as Fastscape difficult. We have tried to make them as similar as possible.

(11) You've switched from deriving stream power model first, then transport model to discussing transport model first, then stream power. It would help readers follow if you kept the same order throughout the paper, but I recognize that's difficult and perhaps not always possible. Methods: I strongly recommend moving Appendix B to the methods section so readers have a better idea what you're doing and what these models look like before discussing results. This is very important and will help the readability of the paper.

We have reorganized the methods to introduce the transport model first. We have also moved Appendix B into the main body of the text.

(12) It seems to me that it's at least intuitively known that aggradational (drying) events happen more slowly than an erosional (wetting) event, but I couldn't point to a reference for this. If there aren't many studies that show this, I think this makes a very interesting point. If there are studies, they should be cited.

We are not aware of any numerical modelling studies which make the explicit point that aggradational drying events are slower than erosional wetting events. If the reviewer has access to studies that show this, we would be very happy to include these in the references.

(13) Can you say more about catchment response time for the sediment transport model? For example, when is this information useful for evaluating sediment records? I think this point needs to be discussed more thoroughly.

We now specify that the transport limited model produces response times of $10^5$ to $10^6$ years. Additionally we state that "To evaluate this question with reference to real examples, we need to consider systems in which the timescales of erosion (or as

a proxy, deposition) are known, stratigraphic sections are complete, and the driving mechanisms well-documented (c.f. Allen et al., 2013; D'Arcy et al., 2017)."

(14) What is the justification for comparing paleosols in the Bighorn Basin with paleosols in Claret? Perhaps too much detail in this section to get to the point of rapid deposition.

We have simplified this section and removed the comparison with the Bighorn Basin. We make use of the Schmitz and Pujalte sedimention constraints as suggested by reviewer 2.

(15) Relevance to sediment record and climate change: This section should include a more generalized framework of the circumstances under which the findings of this study are relevant. What is needed in the field/sedimentary stratigraphy to distinguish between a landscape that was depositing under a detachment-limited vs. a transport-limited system? For example, magnitude and duration of precipitation change, magnitude of sediment flux. It's also important to include an expanded discussion of why the sedimentary record at Claret is difficult to explain with an advective model. How much does precipitation have to change in the time period of deposition for an advective model to work? Is this anywhere close to reality? Generally, I want more to back up the conclusion that the transport model explains deposition in the sedimentary basins simply because it has a faster response time.

We now write "Erosional source catchment areas were likely 100 km in length at the time, given the palaeo-geography of the Pyrenees at the time (Manners et al., 2013). The very short duration of the erosional response, which is required for the sediments to be transported and deposited in a timescale of ca. 10ˆ4 years is therefore difficult to model within an advective end-member model for catchments of this scale, although a version of such a model has been recently used to explore the controls on the evolution of later Miocene megafans in the northern Pyrenees (e.g. Mouchene et al. 2017). (e.g. Table 2). To do so would require us to increase the bedrock erodibility parame-
ter, k, significantly within the model (by greater than one order of magnitude), implying slopes and topography in the palaeo-Pyrenees that were highly subdued indeed. In contrast, the sediment transport model more easily reproduces the documented response timescales given an increase in precipitation, is consistent with the volumetrically significant export of bedload transported gravel clasts, and therefore honors the independent field data more effectively. We also note that the transport model displays a response time that has a stronger dependence on precipitation rate change (e.g. Figure 9). We therefore suggest the erosional pulse that led to the deposition of the Claret conglomerate is most appropriately modelled as a diffusive system response to a sharp increase in precipitation over the source catchments of the developing Pyrenean mountain chain at that time."

However it is important to stress that our model actually only deals with the erosional part of the catchment, and it does not address whether any erosional signal is "sampled" into stratigraphy. This is beyond the scope of this contribution (it is an important question in its own right!) and we acknowledge this on p24 of the manuscript.

Finally, we have addressed all the other minor points raised in this review, and would like to thank the reviewer again for going through our work in such detail.

---

## Referee Report (RR1)

**Review of Numerical modelling landscape and sediment flux response to precipitation rate change, by Armitage et al**

This paper deals with the response of sedimentary systems to sudden increase or decrease in precipitation rate, in terms of timing and magnitude. Starting from the very basic equations of sediment transport, the authors derive two equations that are solve numerically. These equations represent two end-members of the expected dynamics of sedimentary systems. One considers that sediments are immediately evacuated once produced (this is the stream power equation) and the other one considers that there is always some sediments (this is the transport model). The authors explore the response of these two models in terms of sediment flux. They show some differences in terms of magnitude and timing and propose an interesting discussion with respect to natural examples (stratigraphic series associated with the PETM in Spain and in the US).

The authors have seriously taken into consideration the comments and suggestions of the first reviews and the manuscript has been greatly improved. The paper is better organized and discussions of the natural cases are much more robust. However, the manuscript could benefit from some clarifications that I propose below. In addition, typos and inconsistencies in names or convention are found throughout the manuscript. Considering that it should not be an issue to tackle these comments, I recommend accepted with minor revisions.

p 5 l 13-20 and p 17 l2 The definition of Qw is unclear, especially compared to the other equation given p 17. Why do you include a width coefficient ? Why do you use two different expressions and how are they related ?

p 6 l 26 consider adding a comment on the implications of this difference (in the context of this work or in general)

p 8 l 4 the reason for co-variation of the parameters is very clear, but a few words on the choice of the values would be welcome (here or later in the manuscript when you give the values)

p 9 l 5-7, table 1, p 17 l 19 and appendix:  theta is named the slope area exponent (p9, Table 1) or the concavity (p17) or the gradient (p10 l 10) and it is given first in negative and then in positive value. For clarity, please use the same convention.

section 3.1 a few words about why the sediment flux responds in this manner (increase or decrease and then back to previous steady state value) would help readers that are not familiar with such models.

p 15 l 9-10 this sentence is a bit strange here because you have already mentioned differences between the two models.

p 17 l 1 is it possible that there is also a dependency with the uplift rate, which is not explored and therefore not seen in this work ?

p 20 l 9 «even accounting for drainage area» this is not clear

p 24 you use deposition as a proxy for erosion. This is a required approximation when working on the sedimentary archives as a proxy for relief evolution. However, deposition is not directly equivalent to sediment flux and there are some underlying hypothesis that should at least be mentioned, if possible discussed (ie, sediment partitioning along the sedimentary and its potential evolution with climatic perturbation).

p 25 l 12-13 I understand the point of this sentence is to highlight the very high sediment flux at the time of Claret deposition but Maastrichtian to end Palaeocene is ~20 Myrs. Your simulations show

that the sediment flux goes back to «normal» value in ~1 My. This sentence appears quite inappropriate and does not support your work.

Minor comments

Subfigure labels a and b are in or out of the figures. This could be harmonized.
For some figures, using the same vertical axis would help to ease the visualization of the difference between the two models

Figure 12 missing inbox caption in subfigure b (although we can assume it is equivalent to subfigure a)

p 3 l 10, p 7 l 9, p 7 l 22 this work explore only precipitations perturbations

Some typos throughout the manuscript, here are the ones I noticed. The authors should check the text very carefully before final submission:

Abstract (2) and (3) within any (1)
use Equation or equation
p 5 l1 k not defined
p 5 l15 h for the Hack exponent -> h is also the sediment thickness
p 6 l19 qw is already defined
p 8 l 1 lienar
p 9 l 8 extra ( )
p 10 l 7 withn
p 10 l 9 realtionship
p 12 caption precipaiton
p 12 l 10 resonable
p13 l 4 cathcment
p 13 l 10 decease
p 13 l18 : The
p 13 l 5 responce
p 17 extra ,
p 20 caption respectfully X2
p 21 l 8 extra a
p 26 l4 Figure 9 (not 10)
p 24 l 17 1 Myrs

---

## Author Response (AR2)

Insitut de Physique du Globe de Paris
Equipe Dynamique des Fluides Géologiques
1 rue Jussieu
Paris
75005

27[th] November 2017

Dear Jean Braun,

Thank you for taking the time to handle our manuscript and provide guidance on how we can improve it.

In response to your major comment, "I did not see [the authors] addressing the important (second) point raised by the second reviewers, that the authors should try to find a better, more general framework to discuss their findings (and not limit their discussion to the PETM and its recording in the Pyrenees as a case example); this would greatly improve the impact of their work", we have re-written the discussion section, which now includes two introductory paragraphs that try to set our findings in a broader context. We then carefully modified and developed section 4.1 to more broadly discuss the response times and impacts predicted by the models, the asymmetry in model response, and discuss the difficulties in comparing real sedimentary records to these simple numerical models. Subsequently in section 4.2 we have tried to discuss the wider implications of the model as well as providing more details for the three PETM scenarios considered here. And finally in the new section 4.3, we try to summarise the model implications as whole, again highlighting the wider implications and generic findings of this work. We trust these additions, which are tracked in detail, will enable readers to see the generic impact of the work.

In response to your second comment, "Personally, I find also that, in many parts of the paper, the discussion of the results is focused on the comparison between two models and how these can be explained from a mathematical point of view. It would be good if the authors could more often come back to the underlying processes, approximations and hypotheses on which the models are based" we hope in our extensively revised discussion we have managed to focus the text more on the processes. In particular that the stream power model might be more relevant if the sedimentary record is predominantly made up of fine grains that are transported as a suspended load. Conversely the transport limited model is more relevant for bed-load deposits. This means that different parts of the sedimentary record may have different response times and record past climatic change differently. Our new section 4.3 now explicitly considers the limitations of the models in the light of the real physical processes occurring.

The revised discussions include page 24, lines 4 to 19; page 25, lines 1 to 4 and 18 to 30; page 26, lines 13 to 27; page 27, lines 1 to 12; page 29; and page 30 lines 1 to 26.

Finally we have updated the abstract to reflect the wider discussion, and highlight some of the key points that had become lost within our original submission.

Please find below our detailed comments to the two reviews in bold text. We have tried to address all the comments and we hope that our revised manuscript is now considerably improved. We attach a *latexdiff* version of the manuscript to the end of the response to reviews.

Yours sincerely,

John Armitage (on behalf of all the authors).
This paper deals with the response of a landscape to a change in precipitation rate in terms of sediment flux and response time. The authors use numerical modelling to explore the behavior of a landscape considering two end-member for the transport law: the stream power law and the sediment transport law. The first one has been widely used and has showed some limitations. It is therefore of high interest to compare this law to another one. The authors propose several simulations that can be compared with each other and show that the two transport laws lead to a similar first-order behavior but with different response time, that can thus be used to discriminate the on-going processes (instantaneous vs diffusive transport). The authors test their models on the Claret Conglomerate, associated to the PETM (a rapid climatic change) and suggefst that this formation is best explained by a diffusive model. The purpose of this work is of interest for many fields of research. The methodology is consistent and the results are nicely supported by several numerical experiments. However, this work would benefit from some rewriting and reorganization to make the purpose of the authors more clear and easier to follow: some paragraphs could be reorganized and/or developed in particular to better highlight the state of the art in the domain (Introduction) or to develop some important aspects of the work (Method).

Please find below my specific review.

INTRODUCTION

p2 l5 responds to tectono-environmental change : The main results of these models should be mentioned here.

**Added information on both tectonic and climatic forcing: p1: "In general terms, numerical studies have found that landscapes typically will recover from a change in tectonic uplift after $10^5$ to $10^6$ years (e.g. Romans et al., 2016). These apparently long response timescales to tectonic perturbations have been supported by field observations of landscapes upstream of active faults (e.g. Whittaker et al., 2007; Cowie et al., 2008, Whittaker and Boulton, 2012), although the precise appropriateness of any time-integrated erosion law to specific field sites is not always easy to establish."**

**Additionally, we note in the introduction:**
**"The response of landscapes and sediment routing systems to a change in the magnitude or timescale of precipitation rates is expected to depend on the long term erosion law implemented (Castelltort and Van den Driessche, 2003; Armitage et al., 2011; 2013). Some numerical modelling studies, based on treating erosion as a length dependent diffusive problem, suggest that landscape responses to a change in rainfall are also on the order of $10^5$ to $10^6$ years, similar to tectonic perturbations, although they produce diagnostically different stratigraphic signatures from the latter (e.g. Armitage et al., 2011). However, other modelling contributions with different assumptions suggest that response times to a precipitation change may be more rapid (Simpson and Castelltort, 2012), although field data sets remain equivocal (see Demoulin et al., 2017 for a recent review)."**

p2 l9 to evaluate how the response time varies as a function of the model forcing: a few words about why this is important and how this will support your initial question would be welcome.

**Added: "We will focus on the transient period of adjustment to a perturbation in precipitation rates, and using end-member numerical models attempt to evaluate how the response time varies as a function of the model forcing. To this end we aim to find the model parameters that generate similar landscape morphologies such that we can subsequently explore how the same end-member models respond to change in precipitation rate. We believe that the results of this study have implications for understanding the responses of landscapes to past change in climate, and could potentially be compared with and tested against further laboratory experiments."**

p2 after l16 in order to highlight the importance of your work, and to better present the general context, the equations of section 2.1 could be presented and discussed here

**We disagree and would rather keep them within the derivations, so that the models can be compared by a non-specialist.**

p3 l2 could you comment briefly the limitations that are suggested by: in principle ?

**No limitations are suggested, rather by "in principle" we were just adding caution because it is never certain that the deposits represent a complete time history of the erosion of the upland catchment. The phrase "in principle" has been deleted to avoid confusion.**

METHODS

p5 l3-4 Some logical transitions between sentences are necessary here.

**Done.**

p5 l13 the value of the exponent m is usually related to the value of n and is therefore not always close to 0.5. You should rather give a range of m value observed in nature and in experimental landscapes (see for example Kirby and Whipple 200, Lague et al 2003, Wobus et al 2006)

**This is if you relate them to the slope-area analysis, but that is not our point here.**

p5 l17 the last sentences could be supported by more references. In addition, the differences between the linear and non-linear cases should be discussed here.

**We don't quite understand what references are being asked for.**

p5 l24-29 not clear, please consider rewriting

**Rewritten.**

p7 (and Appendix B) please specify which model/soft you used to solve Equation 11

**The 2D equations are solved using a model written in Matlab and developed by Guy Simpson. The 1D version is solved using Matlab and code written by the authors. The text has been adjusted to reflect this. We went through the derivation so that the reader would know what PDE is being solved.**

p7 In Table 1, you mention two model sizes but there is only one in page 7. Please correct.

**Done**

p7 l16-19 you use different grids (square vs triangular) and resolutions (number of nodes) for the two models. Can you please comment on these specific choices and the possible implications (or consider adding a paragraph on this topic in the Discussion) ?

**Grid size matters (see Schoorl JM, Sonneveld MPW, Veldkamp A. 2000. Three‑dimensionallandscape process modelling: the effect of DEM resolution. EarthSurface Processes and Landforms 25: 1025–1034), therefore we tried to make the two models comparable. The sediment transport model solves the equations using a finite element approach, and a triangular grid is 2$^{nd}$ order accurate and an appropriate. The stream power model uses a finite difference approach with a rectangular grid, which is also 2$^{nd}$ order accurate and appropriate. It is known that resolution can effect the drainage patters predicted in these sorts of models, where surface water is routed down the steepest slope of descent. However, we focus mainly on the sediment flux signals out, which are not resolution dependent. Furthermore, when comparing model to model, we have attempted to keep the resolutions similar.**

p7 l20-24 (and in Appendix A and B) this paragraph needs to be developed and more precise: why do you consider two different times (5 and 10 Myrs)? by how much is increased or decrease the precipitation rate ? how do you define the values of the different coefficients ? These values are of main importance for your numerical models and should therefore be discussed more extensively (typical values in natural settings, in experiments, implications, etc).

**Two times: because for some models steady state is achieved quite rapidly, while for others it is not. This is now explained in the methods section.**

p8 the last paragraph could be moved to line 20. Also consider a few words about the choice of the output you follow.

**Done.**

RESULTS

The particular value of m/n = 0.5 has been recently tackled by Kwang and Parker (ESurf 2017). They suggest that m/n = 0.5 leads to unrealistic scale invariance. Can you comment on this ?

**We don't think we can, because we don't think this paper made it through review.**

p9 l1 and l 11 the choice of these values must be explained or supported by some references.

**This was the point of Appendix B, which has now been moved into the Methods section after the comments from the second reviewer.**

p10 Please specify wether you extract only one profile for each model or use several profiles

**We don't understand. On page 10 we discuss the response time, which is calculated form the total volume eroded from the whole model domain. No profiles are used.**

p14 l15-20 these sentences are about the amplitude of the response but it is in a section dedicated to

the response time. Please consider moving this paragraph to a more appropriate section.

**Changed the section heading to "Response to different magnitudes of precipitation rate change".**

DISCUSSION

Based on Figure 6, you propose that the response time is shorter for an increase in precipitations than for a decrease and you also show that the response time is related at first order to the precipitation rate (Figure 7). In your examples (Fig. 6), you start from the same initial precipitation rate (1 m/y) and you end with different values. Therefore, the shorter/longer response time be related to this difference in precipitation rates rather than to increase vs decrease. Can you comment on this ? Some simulations with different initial rates but similar final rates would be very interesting. If, even with similar final rates, the response times are still different according to the scenario, it would be very interesting to discuss why.

**We have added a new section exploring how initial precipitation rate impacts the response time. We have found that for the transport model, the response time is sensitive to the initial precipitation rate. However the proportionality does not fit a power law as was found for the relationship between response time and the final precipitation rate. Furthermore, the change in response time is not very large. For the stream power model the response time is not a function of the initial precipitation rate. We believe this difference is in how the transport model responds directly across the whole catchment, so that slopes at the uppermost reaches of the catchment still have a memory of the previous precipitation rate. For the stream power model, the erosion increases bottom-up and so there is no memory of the previous slope and topography.**

In the second part, the authors present the example of the PETM as a rapid change in precipitations and they discuss the timing of the contemporaneous sediment deposits.

1) Rohl et al propose a duration of 170 kyrs for the PETM. Please add some references for the lower value of 90 kyrs.

**We only use the 170 kyr value now, following Rohl.**

2) You assume a constant rate of deposition for these two formations to estimate a duration of deposition. However, average rate of sedimentation are very difficult to estimate and it is a very strong assumption to consider that a conglomerate is deposited at the rate of a paleosol. Therefore, it seems very difficult to consider that the conglomerate account for 1/3 of the total duration or that it is deposited at a rate or 5 10-4 m/y. In addition, based on your simulations, we expect a change in flux while the system is responding to the change in precipitations. One simple and more robust option is to refer to the value proposed in Schmitz and Pujalte, 2007.

**We have also added the Schmitz and Pujalte reference, giving a range of 10 to 50 kyrs, which we believe to be reasonable and robust. We have deleted the comparison to the rates in the Bighorn basin.**

3) Your work is focused on response time and sediment flux but in this natural example, you document a change in the nature of the deposits. Is there any argument to support a change in Qs ?

**Yes. We now write that "These values suggest sedimentation rates of up to 1 mm/yr, and thus elevated sediment fluxes, which if they had been sustained for the duration of the deposition of**

**the Tremp Group (Maastrichtian – end Palaeocene) would have produced >15 km of sediment thickness, an order of magnitude more than actually observed (Cuevas, 1992)."**

FIGURES

Figure 7 this figure is given for the specific value of 0.5. However, you ran some other simulations with different values of m. Do you observe the same behavior for different m values ?

**We have not checked, but as this aspect was covered in Whipple (2001) and Baldwin et al. (2003), we do not believe we need to check this relationship further for other values of m.**

**We have corrected for the minor comments, where they remained.**
General Comments: The objective of this paper was explore end-member models of landscape evolution with the goals of finding how changes in model forcing and model assumptions control model response time and whether changes in sediment flux during a perturbation is diagnostic of the end member models. The motivation for the study was whether the end member models can constrain sediment fluxes to depositional basins and allow the interpretation of past climate signals from the sedimentary record. Their main findings were that transport-limited models have a faster response time to a change in precipitation rate and sediment flux is higher in transport limited models. In my opinion, one of the most interesting parts is the asymmetry in the response times of transport limited models, with a longer response time to a drying event than to a wetting event, but this result was not discussed in detail. Overall, the quality of the paper including the motivating questions, the derivation of model equations, and the explanation of model set up (in appendix B) was good.

I strongly suggest that the material in appendix B be moved to the Methods section. This will really improve the reading experience. The results were adequately explained, but at times the text was dense with much discussion about the results without references to appropriate figures. The weakest section of the paper was the discussion and conclusions. The authors conclude that a sedimentary record of the PETM is best explained by using a transport-limited model because this model has a faster response time. But a faster response time for one model doesn't rule out another model. I would like to see a more robust support of the sediment transport model in making this conclusion. Second, the authors favor the transport-limited model to explain a particular sedimentary record of the PETM because the instantaneous transport of bedload (assumed by stream power model) is not justified for specific case cited in the Pyrenees. But I don't think anyone would argue that stream power model is justified, so this weakens potential impact of paper.

**We have incorporated Appendix B material into the main paper. Some people invert continent scale river long profiles using a stream power model, and here we show that SPL models results in large scale topographies which are similar to transport model simulations. Also, all the work of Sean Willett (an his team) is based on the SPL, including the chi value approach. These models are applied rather indiscriminately, therefore we think the**

**comparison is somewhat valid.**

In the discussion, I think the paper could benefit from a more generalized framework of the circumstances under which the findings of this study are relevant. For example, what is needed in the field/sedimentary stratigraphy to distinguish between a landscape that was depositing under a detachment-limited vs. a transport-limited system? I think that exploring end member models to identify diagnostic characteristics of each in transient state is a worthy goal, but the authors could do a better job articulating how the diagnostic characteristics they have identified are useful in a larger context outside of interpretation of sedimentary records.

**We have rewritten the discussion section to focus on this. We identify the circumstances when one could compare erosional end-member models with field data (i.e. where the response time is independently constrained, sedimentation rates are high etc). We are also much more specific in the limitations and comparisons between the model end-members. We stress firstly that we address the erosional response of a catchment system to a perturbation – whether this is "captured" in stratigraphy is an additional problem. We also make the point that we wish to compare the generic behaviour of the end-member models. One can always tune one or other to specific field observables.**

Specific Comments:

P1L19-21: Conclusion that rapid response in sedimentary basins more easily explained by using transport model for two reasons: 1) this model has a faster response time (Is this a new finding?) and 2) instantaneous transport of bedload (assumed by stream power model) is not justified for specific case cited in the Pyrenees. But I don't think anyone would argue that stream power model is justified, so this weakens potential impact of paper.

**We think the faster response time is a new finding. Furthermore, in the work of Mouchené et al. (2017), a dominantly stream power based erosion model is applied to the Northern Pyrenees.**

P2L6: "a series of experiments": give very brief description of experiments. Also examples of real catchments responding to changes in precipitation would be useful to give readers an broader framing for your work.

**We have added the following text:**
**"In laboratory studies, a series of experiments where granular piles of a length scale of order of centimeters are eroded due to surface water, have demonstrated that a change in precipitation rate leads a period of adjustment of the landscape topography until a new steady-state is achieved (Bonnet & Crave, 2003; Rohais et al., 2011). These experiments use a mixture of granular silica of a mean diameter in between 10 and 20 µm, that is eroded by water released from a fine sprinkler system above. Given the complexity of these experiments, unfortunately there have been insufficient different precipitation rates studied to fully understand how the recovery time-scale varies as a function of precipitation or other parameters."**

P2L15: Can you also point readers to your own modeled examples of landscapes that were created by transport model and stream power model, but are indistinguishable from each other at steady state.

**Not at this point in the paper but we moved Appendix B into the main part of the text, so they will appear later.**

P2L23: Mentioning that river profiles can be inverted to extract climate history at this point seems to bog down the explanation of differences between advective and diffusive dominated systems in transient state.

**OK. Deleted.**

P2L27-31: Transport/diffusive models do not produce knickpoints in transient state. Pointing out that knickpoints occur for reasons other than a transient state in advective-dominated systems doesn't change that, nor does it support the motivation for the study in the following lines (L31-33). That said, I think exploring model end member behavior is a worthy goal.

**Deleted the paragraph.**

P3L7: typo "dirven"

**Fixed.**

P3L7-8: Be more specific here about what experiments show about response time to a perturbation. It's self-evident that there will be a response time for systems to return to steady state.

**The point is that the controls on the response time are not fully captured.**

P3L10: Make it clear that this is the new piece this study adds to the existing body of knowledge.

**We have added: "Sediment flux response times for the advective stream power law have been previously characterised by Whipple (2001) and Baldwin et al. (2003), and for the transport model they have been studied by Armitage et al. (2011) and Armitage et al. (2013), but not systematically or using 2-D models. Furthermore, to our knowledge no comparison between the transport model has been previously made." to the end of the first paragraph.**

P3L20-21: Be clear and consistent throughout the paper with terminology when referring to advective/stream power law/detachment limited and diffusive/sediment transport model/transport limited. These are used interchangeably throughout the manuscript. I recommend explaining the meaning of all three descriptions for each endmember model early in the paper, then using one of the terms for the rest of the paper.

**We now only refer to transport model and stream power model.**

Figure 1: include erosion, E, in the figure.

**Done.**

P4L10: Why ask the question of if mass transport is appropriate at continent scale when this paper doesn't answer that question. It seems to me the paper addresses the question of if advective transport is appropriate for all models with changing boundary conditions.

**Deleted the question.  In this study we are not changing boundary conditions, rather the precipitation rate which impacts the transport coefficients.**

P4L13-14: reference here, e.g. Davy and Lague 2009.

**Such a citation is to us a bit odd when we are pointing out the obvious.**

P5L1-8: This discussion of mass transport in suspension seems unnecessary here and unrelated to the point of the paper.

**Well, we think not. If mass transport is not in suspension then it is along the bed, and not rapid. Therefore the idea of instantaneous mass transport, which is implicit in the derivation, is wrong and hence the model is nonsense.**

P5L12: define physical meaning of coefficient kp, bedrock erodibility, for readers who are not familiar with erosion models.

**Done.**

P5L20: To this point, the derivation is good, easy to follow for the most part, except authors need to define kp, as noted above.

P5L24-26: alpha (precipitation rate, I found later) is not defined here and makes this section very confusing.

**Done.**

P5L27: more specific definition of kw, width coefficient?

**Done.**

P6L1-5: Before you launch into derivation of transport-limited model, give a few more lines to discussing what this means physically, in a natural system. Also here explicitly say this is the transport-limited erosion equation we use.

**We have addressed this comment in the text.**

P6L23: a fourth name/way of describing stream power as a kinematic wave equation. This is obvious to people who are immersed in the world of erosion models, but many readers are not. So again, be careful and consistent in the terminology used to describe the two end member models.

**We have removed the term "kinematic wave equation".**

P6L25: can you reference a figure that shows a migrating knickpoint?

**We can't, despite them being mentioned in the article cited. We have therefore removed the sentence and reference to the migrating knickpoint.**

P6L25-27: mentioning shock wave seems unnecessary and distracting to me. General comment about derivations: there is lots of discussion of various exponents, but I would like to see explanations of the link exponent values with natural systems where possible.

**It might be distracting to mention shock waves, yet we think it is not unimportant. The model parameters are almost impossible to relate to real observables in the natural system; you can relate them but not as a unique parameter value set, rather as a broad parameter value space (see e.g. Croissant and Braun 2014). The models are instead trying to capture some gross simplification of the natural system. A natural manifestation of a shock wave would be the**

**upward migration of a waterfall within the catchment. Of course waterfalls could form due to many other natural circumstances. The point however remains that if we chose to use the stream power law, we should be aware that this model can lead to shock waves.**

P7L16-19: why do you use different grid sizes for the two models? Does this affect the outcome of the models?

**See the reply to the same question from Laure Guerit.**

P7L22-24: This is confusing on the first read through. Explain more or reference figure that shows this relationship (maybe Figures 12 or 14)

**Re-written.**

P7L25: you've switched from deriving stream power model first, then transport model to discussing transport model first, then stream power. It would help readers follow if you kept the same order throughout the paper, but I recognize that's difficult and perhaps not always possible. Methods: I strongly recommend moving Appendix B to the methods section so readers have a better idea what you're doing and what these models look like before discussing results. This is very important and will help the readability of the paper.

**We have re-organised the methods so that the sediment transport model is derived first, and the stream power model second. Appendix B is now also in the Methods section.**

Figures 2 and 3: These figures are not high impact by themselves. Suggest combining these figures and then give them clear titles indicating which end member model results are shown.

**Combined.**

P8L16-17: this was the first time it was clear to me that the heart of this paper is model response to changing precipitation rates. Emphasize this more clearly in the methods section.

**We have tried to explain this better in the text.**

P9L2: Sentence that begins, "The response to a reduction in precipitation. . .." is confusing because it's not specific and is a bit of a run on. Needs punctuation or splitting into two sentences to make it clearer.

**Re-written**

P9L3-6: Very long run on sentence. This sentence says "importantly, however, it changes the model elevation. . .". Reference a figure where readers can see this change in elevation with changing c.

**Re-written**

P9L4: Be specific about where in appendix A readers should look

**Re -written**

P9L10-11: Reference one of your figures that shows where these three different sets of parameters result in similar model topography, e.g. figure 14.

**Done**

Figure 4 and 5: These need titles to make it clear that one is the transport model and one is the stream power model.

**Done**

Figure 5: indicate knickpoint on figure 5a

**The text mentioned knickpoints in reference to the paper by Jean Braun (Braun et al., 2015). In Figure 5 we find it hard to pick a knickpoint, therefore we have removed the phrase "The lower reaches of the catchment respond more rapidly than the upper reaches, therefore creating a migrating knickpoint as the landscape responds to the change in model forcing (see Braun et al., 2015)."**

Figure 6: this figure must have titles indicating that one is the transport model and one is the stream power model.

**Done**

P12L3-5: It seems to me that it's at least intuitively known that aggradational (drying) events happen more slowly than an erosional (wetting) event, but I couldn't point to a reference for this. If there aren't many studies that show this, I think this makes a very interesting point. If there are studies, they should be cited.

**We are not aware of any numerical modelling studies which make the explicit point that aggradational drying events are slower than erosional wetting events.  If the reviewer has access to studies that show this, we would be very happy to include these in the references.**

Figure 7: include in caption these results apply to catchments where L=100km only. Should also make clear where the data to make this figure comes from. Is it just a line through your model data? If made from model data, these should be indicated with points and justification given for how the line was drawn through the model data.

**Done.**

P13L1-2: include reference for knickpoint celerity models, e.g. Berlin and Anderson 2009, JGR

**Why? We were not making any point about knickpoint celerity, unless we are not understanding something here.**

P14L2-3: This seems important that response time takes twice as long when L=500 km compared to L=100km. I suggest a figure showing this difference in response time.

**A figure has been added.**

P14L12-13: explain what this empirical evidence of 0.5<h<0.7 means in a physical system.

**Added that this controls the plan view shape of river catchments, but as far as we are aware there is still no explanation for the origin of this scaling (see Dodds & Rothman, "Geometry of river networks. I. Scaling, fluctuations, and deviations", Physical Review E, 2000).**

Figure 9: show points where you have data from model runs.

**Done**

P16L2-3: reference figure 7 that shows how models respond to precipitation rate.

**Done**

P16L6: reference figure that shows similarities between landscapes created by transport models and advective models.

**Added figure references.**

P16L7-9: I appreciate the goal, but what constraints are needed to make this useful? See comment below on conclusions.

P16L13: reference figure that shows difference in response times.

**Done**

P17L1-2: this asymmetry is interesting. Reference figure that shows it.

**Done**

P17L7-8: Can you say more about catchment response time for the sediment transport model? For example, when is this information useful for evaluating sediment records? I think this point needs to be discussed more thoroughly.

**We have added a section on the "Relevance of model responses to sediment records of climate change" where we now hope these points are discussed more thoroughly.**

P18L2: At Claret: mention this is the site in the Pyrenees.

**Done**

P18L6-10: confusing. What is the justification for comparing paleosols in the Bighorn Basin with paleosols in Claret? Perhaps too much detail in this section to get to the point of rapid deposition.

**We have eliminated this comparison. We now provide two estimates of sedimentation rate for the Claret conglomerate (including the low end-member estimate of 10 kyrs from Schmitz and Pujalte 2007). We adopt a timescale of ca. 200 kyrs based on the work of Brady Foreman for comparison with US PETM sites.**

P19L12-14: I don't see why one would attempt explain evolution of a megafan with a stream power model in the first place. This is a bit of a strawman argument.

**Actually very recently the large megafans of the Northern Pyrenees have been modelled using a model where erosion is calculated using the stream power model (Mouchene et al. 2017). Furthermore, the key point is not the evolution of the mega fan – we are actually talking about how the eroding catchment which is producing the sediment is modelled. One could model catchment erosion using a stream model upstream of a fault and have a sensible sediment flux prediction – yet still want to use this sediment flux model as an input to a**

**depositional stratigraphic model.  To make sure the reviewer and readers don't mix these two points, we repeatedly clarify in the discussion that we are talking about the erosional part of the catchment.**

P18L20-24: Run on sentence.

P19L1-3: Again, it seems obvious that bed load transport is more easily described by a diffusive/transport limited model rather than the stream power model. It would be useful to reference studies where authors have used stream power to model bedload dominated systems.

**We reference the Mouchene paper, mentioned in the response above.  The point is again, that numerical models can produce sediment flux predictions based on a stream power model. These sediment flux predictions could be used to build stratigraphy in a subsiding basin.  We stress the that it is important to separate out erosion and deposition when considering these systems.**

 Relevance to sediment record and climate change: This section should include a more generalized framework of the circumstances under which the findings of this study are relevant. What is needed in the field/sedimentary stratigraphy to distinguish between a landscape that was depositing under a detachment-limited vs. a transport-limited system? For example, magnitude and duration of precipitation change, magnitude of sediment flux. It's also important to include an expanded discussion of why the sedimentary record at Claret is difficult to explain with an advective model. How much does precipitation have to change in the time period of deposition for an advective model to work? Is this anywhere close to reality? Generally, I want more to back up the conclusion that the transport model explains deposition in the sedimentary basins simply because it has a faster response time.

**We now write:**

**Erosional source catchment areas were likely < 100 km in length, given the palaeo-geography of the Pyrenees at the time (Manners et al., 2013). The very short duration of the erosional response, which is required for the sediments to be transported and deposited in a timescale of ca. 10 $^4$ years is therefore difficult to model within a stream power (advective) end-member model for catchments of this scale (Table 2), although a version of such a model has been recently used to explore the controls on the evolution of later Miocene megafans in the northern Pyrenees (e.g. Mouchené et al., 2017). To use the stream power model would require a significant increase the bedrock erodibility parameter, *k*, within the model (by greater than one order of magnitude), implying slopes and topography in the palaeo-Pyrenees that were highly subdued. In contrast, the sediment transport model more easily reproduces the documented response timescales given an increase in precipitation, is consistent with the volumetrically significant export of bedload transported gravel clasts, and therefore honors the independent field data more effectively. We also note that the transport model displays a response time that has a stronger dependence on precipitation rate change, and has a greater amplitude of perturbation (e.g. Figure 9). We therefore suggest the erosional pulse that led to the deposition of the Claret conglomerate is most appropriately modelled as a diffusive system response to a sharp increase in precipitation over the source catchments of the developing Pyrenean mountain chain at that time."**

**However it is important to stress that our model actually only deals with the erosional part of the catchment, and it does not address whether any erosional signal is "sampled" into stratigraphy.  This is beyond the scope of this contribution (it is an important question in its own right!) and we acknowledge this on p24 of the manuscript.**

P19L10-12: I don't think that noting that knickpoints are not a unique indicator of erosional dynamics is helpful for understanding the motivation of this study.

**We would prefer to leave this here as the point about knickpoints not being unique is useful because just because you see one in a river it does mean you should leap for a detachment limited stream power law.**

P19L23-24: This is one of the most interesting outcomes of the paper. I would suggest discussing this and the implications of this effect on interpretation of the sedimentary record.

**Good suggestion. We have added at the end of the discussion that "Nonetheless, a significant finding of this work has been the clear asymmetry in response time of these end-member models in terms of a wetting event (faster) compared to a drying event (slower). This implies that aridification events are harder to preserve in the sedimentary record, not only because they are typically associated with reduced sediment fluxes, but also because the timescale of landscape response may be > 10^6 years".**

P20L1-4: nicely summarized objective of the paper. I would like to see this in the abstract also
Figure 10: Figure needs a legend for the lines

**Done.**

Figure 11: It's not immediately clear why both Figure 10 and Figure 11 are necessary. It's explained further down, P22L3-7, but figure 11 is initially confusing.

P22L12-13: show example of how slope-area relationship is sensitive to river network in T-Lim case. Or remove this line as it's not relevant.

**Deleted.**

P22L14-20: The point of this paragraph is unclear even after reading several times. Rewrite and reword.

**Re-written.**

Appendix B: As noted above, I think appendix B should be included before results are discussed. It would make the paper flow much more smoothly.

**Done.**

Figure 12: Labels on the figures: transport limited models.

**Done.**

P24L5: typo, should read steady state.

Figure 14: Figure needs label: stream power models. Also, there is an error in the caption. The caption currently says sediment transport model, should read stream power model.

**Done.**

[revised manuscript text omitted]
 = 10^{-2}\,(\text{m}^2\,\text{yr}^{-1})^{1-\delta}$, $\delta = 1.3$ with $c = 10^{-3}\,(\text{m}^2\,\text{yr}^{-1})^{1-\delta}$, and $\delta = 1.5$ with $c = 10^{-4}\,(\text{m}^2\,\text{yr}^{-1})^{1-\delta}$ (Figure 2b).

Sediment flux out of the model domain for the sediment transport model for models where (a) $\delta = 1.1$, $1.3$ and $1.5$, $\kappa = 10^{-2}$ and $c = 10^{-4}\,(\text{m}^2\,\text{yr}^{-1})^{1-\delta}$, and (b) $\delta = 1.1$, $1.3$ and $1.5$, $\kappa = 10^{-2}$ and $c = 10^{-2}$, $10^{-3}$, and $10^{-4}\,(\text{m}^2\,\text{yr}^{-1})^{1-\delta}$.

When the transport coefficient $c$ is the same for the three values of the exponent $\delta$ the model wind-up time increases with decreasing $\delta$, and takes several million years where $\delta < 1.5$ (Figure 2a). Steady state sediment flux is greater for increasing $\delta$ when $c$ is kept constant. The dimensions (units) of $c$ depend on $\delta$ which means that the value of the coefficient $c$ must be adjusted when $\delta$ is changed to yield the same unit erosion rate per water flux, regardless of $\delta$ (see Armitage et al., 2013). Consequently, when $c$ is suitably adjusted the model can reach a steady state in a similar time for all three values of $\delta$ (Figure 2b).

(a) Steady state topography, after 10Myr, for the sediment transport model where $\delta = 1.5$ and $c = 10^{-4}\,(\text{m}^2\,\text{yr}^{-1})^{1-\delta}$. (b) Slope area relationship for sediment transport model for $\delta = 1.3$ and $\delta = 1.5$.

We subsequently analyze the topography for the relationship between trunk river slope and drainage area, Figure 3, using Topotoolbox2 (Schwanghart and Scherler, 2014). For the case where $\delta = 1.5$ the scaling between channel slopes and catchment drainage areas, slope area gradient, $\beta$ is equal to -0.42, and for $\delta = 1.3$, $\beta$ is equal to -0.23 (Figure 3b). The same value is calculated using the spatial transformation described within (Perron and Royden, 2012), commonly referred to as $\chi$-plots (Table 1). Given the reduction in $\beta$ from $\delta = 1.5$ to $1.3$, we did not analyze the case for $\delta = 1.1$ as the slope-area relationship will clearly lie below the observed range ($0.3 < \beta < 0.7$; e.g. Whipple and Tucker, 2002; Tucker and Whipple, 2002). Therefore, for river networks defined by routing water down the steepest slope of descent, the sediment transport model can create catchment morphologies that have a concavity similar to that observed in nature if $\delta \sim 1.5$.

Slope area relationship for trunk streams sediment transport $k_S$ $\beta$ $\delta = 1.3$ 0.86 -0.23 $\delta = 1.5$ 1.76 -0.42 stream power $m = 0.3$ 0.95 -0.29 $m = 0.5$ 6.52 -0.46 $m = 0.7$ 71.42 -0.68

**A1  Comparison to Erosion by Stream Power**

Sediment flux out of the model domain for the sediment transport model for models where (a) $m = 0.3$, $0.5$ and $0.7$, and $k = 10^{-5}\,(\text{m}^2\,\text{yr}^{-1})^{1-m}$, and (b) $m = 0.3$, $0.5$ and $0.7$, and $k = 10^{-4}$, $10^{-5}$, and $10^{-6}\,\text{m}^{-1}\,(\text{m}^2\,\text{yr}^{-1})^{1-m}$.

In order to provide a comparison for the morphology of the sediment transport model we return to the widley used stream power model. We explore how the stream power model evolves to a stead state for a range for the coefficient $k$ and the exponent $m$.

The landscape derived from the stream power model, equation 11, evolves towards a steady state with a slightly different behaviour in comparison to the sediment transport model (Figure 4). As before we run six models where in this case the first set of three are $m = 0.3$, $0.5$ and $0.7$ with $k = 10^{-5}\,\text{m}^{-1}\,(\text{m}^2\,\text{yr}^{-1})^{1-m}$ (Figure 4a). The second set of three are of $m = 0.3$ with $k = 10^{-4}\,\text{m}^{-1}\,(\text{m}^2\,\text{yr}^{-1})^{1-m}$, $m = 0.5$ with $k = 10^{-5}\,\text{m}^{-1}\,(\text{m}^2\,\text{yr}^{-1})^{1-m}$, and $m = 0.7$ with $k = 10^{-6}\,\text{m}^{-1}\,(\text{m}^2\,\text{
[revised manuscript text omitted]

---

## Author Response (AR3)

Institut de Physique du Globe de Paris
1 rue Jussieu
Paris, 75005
France

9 January 2018

Dear Jean Braun,

We would like to thank you for allowing us the opportunity to respond to Laure Geurit's review, and would like to take this opportunity to thank Laure for her thorough and very helpful reviews. We have tried to incorporate all of her comments into this revised submission. Below we respond and outline what we have done to each specific comment, and attach a version of the manuscript with the changes tracked.

Yours sincerely
John Armitage

(on behalf of all of the authors)

–

Responses (in bold text) to review comments by Laure Geurit:

p 5 l 13-20 and p 17 l2 The definition of Qw is unclear, especially compared to the other equation given p 17. Why do you include a width coefficient ? Why do you use two different expressions and how are they related ?

**We have to include a coefficient to make the units of sediment flux consistent. Length is raised to the power $p$, while water flux is in units $m^2$/yr. On page 17, the phrase was badly written, and on reflection served no purpose, so has been deleted.**

p 6 l 26 consider adding a comment on the implications of this difference (in the context of this work or in general)

**We have added the following, line 26:**
**"Furthermore if n > 1, equation 11 becomes non-linear and the model response to precipitation rate change will become a function of both uplift and precipitation rates (Whipple, 2001)."**

p 8 l 4 the reason for co-variation of the parameters is very clear, but a few words on the choice of the values would be welcome (here or later in the manuscript when you give the values)

**We have added the follwing, line 31:**
**"These values are chosen because they generate response times within the range of observations from normal fault bounded sedimentary systems that have responded to changes in slip rate (Densmore et al., 2007; Armitage et al., 2011)."**

p 9 l 5-7, table 1, p 17 l 19 and appendix: theta is named the slope area exponent (p9, Table 1) or the concavity (p17) or the gradient (p10 l 10) and it is given first in negative and then in positive value. For clarity, please use the same convention.

**We have corrected this inconsistency and chosen "slope area exponent" and kept the values negative.**

section 3.1 a few words about why the sediment flux responds in this manner (increase or decrease and then back to previous steady state value) would help readers that are not familiar with such models.

**We have added, page 12, line 12:**
**"When the model is perturbed by a change in precipitation rate the sediment flux output will first change, as the erosive power changes (e.g. Figure 6). The model will subsequently return to the steady-state output, as the slope of the fluvial system will adjust to the new precipitation rate, and the landscape will re-achieve the same steady-state."**

p 15 l 9-10 this sentence is a bit strange here because you have already mentioned differences between the two models.

**We have removed the sentence.**

p 17 l 1 is it possible that there is also a dependency with the uplift rate, which is not explored and therefore not seen in this work ?

**In a previous study, Armitage et al. (2013), we found that for coupled erosion deposition models, the uplift rate did not strongly affect response times. For the linear model discussed on page 17, uplift should not impact response times, however when non-linearities are included, like slope exponents, then uplift will impact response times. It is for this reason that we subsequently investigated the effect of uplift.**

p 20 l 9 «even accounting for drainage area» this is not clear

**We agree and have removed it.**

p 24 you use deposition as a proxy for erosion. This is a required approximation when working on the sedimentary archives as a proxy for relief evolution. However, deposition is not directly equivalent to sediment flux and there are some underlying hypothesis that should at least be mentioned, if possible discussed (ie, sediment partitioning along the sedimentary and its potential evolution with climatic perturbation).

**We have added the following, page 24, line 21:**
**"To compare our model predictions with observations it is clear that we have to use the depositional record. Therefore, there is an implicit assumption that stratigraphy is a faithful recorder of erosion. It is however possible that climatic change will also alter processes that control sediment deposition, for example by altering how sediment partitions from transport into stratigraphy. By using estimates of the total volume of sediment deposited within the Escanillia Eocene sedimentary system in the Spanish Pyrenees, it has been demonstrated that climatic change can recreate observed change in grain size deposition (Armitage et al., 2015). This example of a close model-to-stratigraphic-observation prediction might be evidence that the stratigraphic record is a faithful record of change in sediment flux delivery to the depositional environment."**

p 25 l 12-13 I understand the point of this sentence is to highlight the very high sediment flux at the time of Claret deposition but Maastrichtian to end Palaeocene is ~20 Myrs. Your simulations show that the sediment flux goes back to «normal» value in ~1 My. This sentence appears quite

inappropriate and does not support your work.

**Our point was that the sedimentation rate from stratigraphic constraints before the PETM was a lot lower than the very high rates during the PETM event - showing the sudden increase at that time. This was obviously not clearly explained, we have therefore re-written these sentences. The text is now the following, page 25, line 24:**

[revised manuscript text omitted]